# Adaptive Gradient Masking for Balancing ID and MLLM-based Representations in Recommendation

**Yidong Wu** *
Imperial College London
London, UK

**Siyuan Chen** *
University of Bristol
Bristol, UK

**Binrui Wu**
Fudan University
Shanghai, China

**Fan Li**
KuaiShou Technology
Beijing, China

**Jiechao Gao** †
Stanford University
Stanford, USA

## Abstract

In large-scale recommendation systems, multimodal (MM) content is increasingly introduced to enhance the generalization of ID features. The rise of Multimodal Large Language Models (MLLMs) enables the construction of unified user and item representations. However, the semantic distribution gap between MM and ID representations leads to *convergence inconsistency* during joint training: the ID branch converges quickly, while the MM branch requires more epochs, thus limiting overall performance. To address this, we propose a two-stage framework including MM representation learning and joint training optimization. First, we fine-tune the MLLM to generate unified user and item representations, and introduce collaborative signals by post-aligning user ID representations to alleviate semantic differences. Then, we propose an Adaptive Gradient Masking (AGM) training strategy to dynamically regulate parameter updates between ID and MLLM branches. AGM estimates the contribution of each representation with mutual information, and applies non-uniform gradient masking at the sub-network level to balance optimization. We provide theoretical analysis of AGM's effectiveness and further introduce an unbiased variant, AGM*, to enhance training stability. Experiments on offline and online A/B tests validate the effectiveness of our approach in mitigating convergence inconsistency and improving performance.

## 1 Introduction

Large-scale industrial recommendation systems have traditionally relied on ID-based features, such as cross ID features (e.g., FM [1], DCN [2]), list-wise ID features (e.g., DIN [3], TWIN [4]), to model user–item interactions. These features offer strong memorization capabilities and are effective in capturing co-occurrence patterns. However, they suffer from limited generalization, making them inadequate for addressing long-tail items, data sparsity, and cold-start scenarios. To mitigate these limitations, recent research [5, 6, 7] has incorporated multimodal (MM) content (e.g., images and textual descriptions) to enrich the semantic representations of users and items.

The emergence of Multimodal Large Language Models (MLLMs) facilitates the generation of unified, high-level semantic embeddings from diverse modalities, offering promising avenues for enhancing recommendation performance. As is demonstrated in many recent studies, integrating categorical features (e.g., ID and category) with MLLM-based representations (e.g., images and texts) can

---

*Equal Contribution
†Corresponding Author: `jiechao@stanford.edu`

39th Conference on Neural Information Processing Systems (NeurIPS 2025).

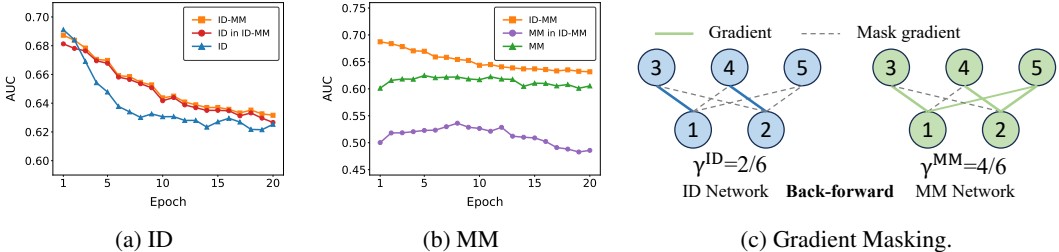

(a) ID      (b) MM      (c) Gradient Masking.

Figure 1: (a) Validation-set AUC comparison between the ID-only model (ID) and the combined model (ID-MM), where "ID in ID-MM" denotes the contribution of the ID component within the combined model; (b) Validation-set AUC comparison between the multimodal-only model (MM) and the combined model (ID-MM), where "MM in ID-MM" denotes the contribution of the multimodal component within the combined model. (c) The illustration of Adaptive Gradient Masking.

effectively foster the performance of recommendation system [8, 9, 10, 11]. Works like FREEDOM [9] introduce auxiliary losses on multimodal data to refine ID embeddings through graph structures. Recent self-supervised learning models such as BM3 [10] adopt joint training of content-based and user-item objectives. AlignRec [12] proposes a unified alignment framework that aligns visual, textual, and categorical modalities using pretraining and contrastive learning.

Despite these advances, integrating ID and MLLM-based representations remains challenging due to convergence inconsistency during joint training. This stems from a semantic gap—ID embeddings encode co-occurrence patterns, while MLLM features capture high-level semantics—and an optimization imbalance, as ID embeddings are trainable whereas MLLM parameters are typically frozen. As a result, the ID branch quickly dominates training, leading to biased gradients that suppress the multimodal branch and ultimately degrade overall performance. To address this issue, although AlignRec [12] alleviates this convergence speed mismatch between multimodal and ID features by adopting a two-stage strategy: first, pre-training the alignment of content modes and then performing joint training, it does not fundamentally address the imbalance during the joint optimization phase. In contrast, we take a more fine-grained approach by dynamically balancing gradient updates between ID and multimodal branches.

In more detail, we propose a two-stage framework consisting of multimodal representation learning and joint training optimization. Firstly, we fine-tune a pretrained Multimodal Large Language Model (MLLM) to generate unified multimodal representations for items and users from visual and textual content. Specifically, Item representations are obtained from multimodal prompts through a designated output token, while user representations are derived by aggregating historical item representations. Besides, to bridge the semantic gap between ID and MM representations, we introduce collaborative alignment by post-aligning multimodal embeddings with their corresponding ID embeddings. In the second stage, we jointly train the recommendation model by combining ID and MLLM-based representations under the binary cross-entropy objective, and propose an Adaptive Gradient Masking (AGM) strategy to dynamically regulate their parameter updates during optimization. AGM estimates the informativeness of each representation through mutual information, and applies non-uniform gradient masking at the subnetwork level to encourage balanced convergence. This adaptive mechanism prevents the ID branch from dominating training and ensures consistent convergence across both branches. To demonstrate the effectiveness of our approach, we provide a theoretical analysis showing that AGM leads to more balanced gradient updates, thereby promoting consistent convergence of both branches. Furthermore, we propose an unbiased variant, AGM*, which improves training stability by correcting the bias introduced by binary masking.

Our contributions are summarized as follows:

- We analyze *convergence inconsistency* issue between ID and MLLM-based representations during joint training in recommendations, and propose a two-stage method to address it.

- We propose Adaptive Gradient Masking (AGM), a subnetwork-level optimization strategy that dynamically balances gradient updates between ID and multimodal branches. We also introduce AGM*, an unbiased variant that enhances stability.

- We fine-tune MLLMs to generate unified user and item representations, and introduce a collaborative alignment mechanism to bridge the semantic gap between ID and MM embeddings.
- We provide theoretical analysis on the convergence properties of our method, and validate their effectiveness through extensive offline experiments and online A/B testing in a large-scale industrial recommendation system.

## 2  Related works

**Multi-modal recommendation**   Multi-modal recommendation extends the classic collaborative filtering paradigm by integrating diverse content modalities (e.g., images, text, videos) to capture richer contextual signals and thereby enhance recommendation accuracy. Early approaches, such as VBPR [13] and methods fusing visual features with ID embeddings [14, 15], showed that combining basic item side information can significantly improve user–item matching. Subsequently, attention-based architectures, including VECF [16] and MAML [17], explored finer-grained user preferences by leveraging mechanisms like image segmentation [18] and multi-modal feature interactions. With the surge of Graph Neural Networks (GNNs) in recommendation [19], models like MMGCN [20], GRCN [21], and DualGNN [22] pushed multi-modal recommendation further by injecting high-order neighbor relationships or user attentions across item modalities into node representations. To better reveal item–item semantic similarities, LATTICE [23] constructs separate item–item graphs for each modality and fuses them into a latent graph, while MVGAE [24] employs a modality-specific variational graph autoencoder to combine multi-modal embeddings. Later, MGCN [8] constructs separate graph views to fuse text, image, and user–item interactions more effectively. Recent work like GUME [25] focuses on leveraging semantic neighbors and refining user modality embeddings to strengthen long-tail item connectivity, while LGMRec [26] separates local user–item interactions from global attribute relationships via hypergraph modeling. FREEDOM [9] tackles noisy item–item structures by freezing precomputed graphs and pruning user–item edges. Self-supervised learning method such as BM3 [10] proposes a self-supervised learning framework that relies on latent embedding dropout to create view augmentations. AlignRec [12] addresses alignment challenges across different modalities by unifying multi-modal content and ID-based features through a multi-stage alignment process. In addition, the remarkable progress of foundation models in various modalities [27, 28, 29] has prompted researchers to adopt large-scale pretrained encoders for capturing more holistic multi-modal representations. Typical examples include VIP5 [30], which extends the text-based P5 [31] by incorporating a CLIP image encoder, and MMGRec [32], which reveals item IDs from both collaborative and multi-modal signals via a Graph RQ-VAE. Moreover, IISAN [33] proposes a lightweight Decoupled PEFT architecture that simultaneously tackles intra- and inter-modal adaptation in a plug-and-play manner.

**Multi-modal Large Language Model**   Multi-modal Large Language Models (MLLMs) have recently achieved significant progress in integrating language with other modalities, driven by the surge in large-scale pretraining [29, 34, 35, 36]. Research efforts generally begin with multimodal understanding and text generation, with representative models such as BLIP-2 [37] and LLAVA [38]. Models like LLaMA-Adapter [39, 40] and mPLUG-Owl [41, 42] align text and image features via extensive image–text pairs, while InstructBLIP [43] reshapes multiple tasks into instruction-based formats. Despite such progress, enhancing the visual encoder resolution [44, 45, 46, 47, 48, 49] can result in prohibitive memory overhead, especially in multi-page scenarios. To address such a problem, TextMonkey [50] employs token resampling to reduce the visual token load. Similarly, more recent models such as Qwen2-VL [51] and GPT-4 [52] have exhibited outstanding proficiency in multimodal reasoning and generation.

## 3  Methodology

### 3.1  Problem formulation

We consider the Click-through Rate (CTR) task defined on a dataset $\mathcal{D} = \{(u_i, v_i, y_i)\}_{i=1}^{N}$, where each sample consists of a user $u = (\mathbf{e}_u^{\mathrm{id}}, \mathbf{e}_u^{\mathrm{mm}})$, an item $v = (\mathbf{e}_v^{\mathrm{id}}, \mathbf{e}_v^{\mathrm{mm}})$, and a binary label $y \in \{0, 1\}$ indicating whether the user engaged with the item. Here, $\mathbf{e}_u^{\mathrm{id}}$ and $\mathbf{e}_v^{\mathrm{id}}$ denote trainable ID embeddings

for user and item, respectively, while $\mathbf{e}_u^{mm}$ and $\mathbf{e}_v^{mm}$ represent multi-modal representations of user and item, extracted from MLLM. The goal is to learn a prediction function $f(u_i, v_i; \boldsymbol{\theta})$ that estimates the probability of user–item interaction. The model is trained to minimize binary cross-entropy loss.

## 3.2 Multimodal representation learning

Multimodal information, such as text, images, and other item-related metadata, provides substantial advantages by enhancing the representation of both items and users in recommendation systems. However, although pre-trained Multimodal Large Language Models (MLLMs) excel at understanding the data representation [52, 53, 54], their original evaluation metrics are not specifically designed to meet the unique demands of recommendation tasks. As a result, when faced with extensive user and item information, these pre-trained models often struggle to extract key, effective feature embeddings. To address this limitation and improve the extraction of multimodal features, we introduce a novel approach to efficiently generate task-relevant embeddings by harnessing multimodal features.

### 3.2.1 Item embedding

In this section, we fine-tune the MLLM using three novel alignment objectives aimed at enhancing cross-modal consistency. Furthermore, we append a special token, [Item_cls], to the end of each item description, which allows the model to condense lengthy multimodal token sequences into compact and informative embeddings.

For item $i$, we first combine its textual and visual attributes into a unified input description. This is accomplished using a specific prompt template designed to guide the model's multimodal understanding: "Integrate text and visual information into an embedding representation. Textual:`[Text]`, Visual: `[Image/video]`." Then the MLLM encodes the input and generates a corresponding token sequence including [item_cls], in the form of $\{t_1, t_2, ..., t_m, [item\_cls]\}$. Finally, the hidden state associated with the [Item_cls] token is extracted as the multimodal embedding for item $i$.

$$\mathbf{e}_v^{mm} = \text{MLLM}(\text{text}_i, \text{image}_i), \tag{1}$$

where $\mathbf{e}_v^{mm}$ donates the multimodal item embedding of item v.

In the fine-tuning phase, we introduce three specialized alignment tasks for multimodal recommendation, aimed at improving the MLLM's performance and suitability in recommendation scenarios.

**Text-image alignment**: To align visual and textual features, we adopt a method inspired by BERT [55]. For item $i$ with image $V_i$ and text $T_i$, we mask 20% of $T_i$'s tokens with a special `[MASK]` token, obtaining $\hat{T}_i$. The model then takes $(V_i, \hat{T}_i)$ as input, with the corresponding original description $T_i$ as the target output. This reconstruction task compels the model to leverage visual information to infer missing textual content, thereby learning the meaningful relationship between visual features and textual context for improved cross-modal understanding.

**Meta-data processing**: Recommendation systems leverage both structured metadata (e.g., title, price, tags) and unstructured descriptions. Since metadata directly reflects item characteristics, its effective processing enhances MLLMs' encoding performance. Thus, for item $i$, we propose predicting its detailed description $T_i$ from its metadata, establishing a robust mapping between structured attributes and unstructured text.

**User behavior understanding**: The model explicitly captures interest evolution patterns by predicting users' future interactions based on their multimodal historical behavior sequences, enabling adaptive optimization of recommendation strategies. For this purpose, we create fine-tuning samples where a user's interaction history (containing both textual and visual item features) serves as input, while the next interacted item provides the supervision signal.

### 3.2.2 User embedding

Despite the basic user information, analyzing historical item sequences is also crucial for predicting user preferences. However, handling extensive user histories and aligning textual information with corresponding images presents a significant challenge in multimodal recommendation scenarios. To address the efficient aggregation of long multimodal sequences, we propose the User Embedding Generator (UEG). This module is designed to efficiently aggregate the sequence multimodal information

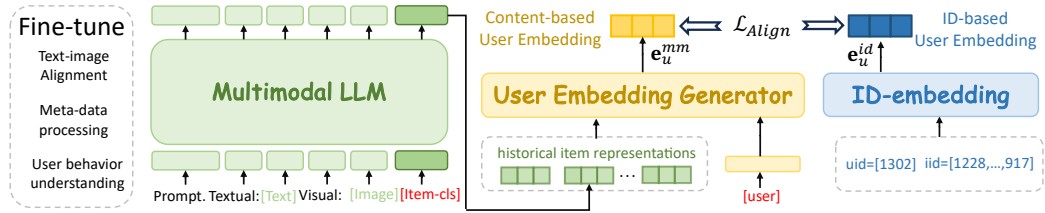

Figure 2: Multimodal representation learning

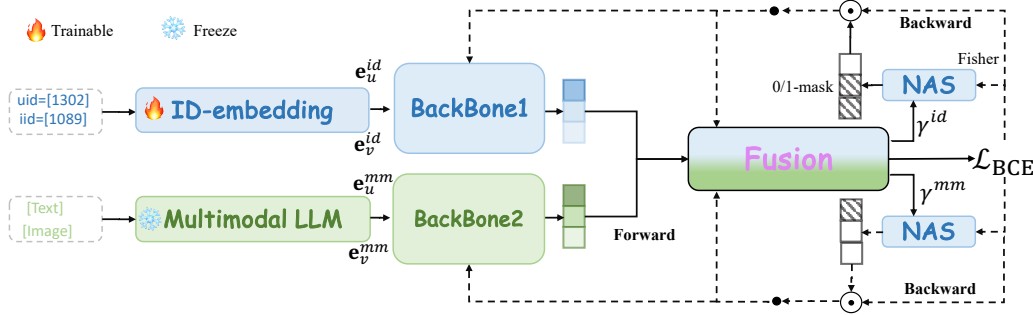

Figure 3: Adaptive Gradient Masking(AGM).

of user historical items into a unified user multimodal representation. For user $u$,

$$\mathbf{e}_u^{mm} = \mathrm{UEG}(\mathbf{e}_{v_1}^{mm}, \mathbf{e}_{v_2}^{mm}, ..., \mathbf{e}_{v_s}^{mm}) \tag{2}$$

As illustrated in Fig. 2, the UEG is a learnable neural network module that takes both the multimodal representations of a user's historical interactions and their unique identifier as input, producing a comprehensive multimodal user embedding $\mathbf{e}_u^{mm}$. To stabilize the learning of multimodal user representations, we further incorporate a pre-trained ID-based embedding layer(with frozen parameters), which generates an ID-based user representation $\mathbf{e}_u^{id}$ from the user's identifier and historical interactions. The UEG module is optimized via an alignment loss:

$$\mathcal{L}_{align} = \|\mathbf{e}_u^{id} - \mathbf{e}_u^{mm}\|_2^2 \tag{3}$$

This objective ensures the learned multimodal representations maintain consistency with established ID-based embeddings while capturing rich multimodal patterns.

### 3.3 AGM

**Forward propagation and convergence inconsistency** Before computing interaction logits, we first obtain ID-based embeddings $(\mathbf{e}_u^{id}, \mathbf{e}_v^{id})$ through trainable embedding layers and multimodal embeddings $(\mathbf{e}_u^{mm}, \mathbf{e}_v^{mm})$ using the method described in Section 3.2. These are processed through separate backbones $g^{id}(\cdot, \cdot; \boldsymbol{\theta}^{id})$ and $g^{mm}(\cdot, \cdot; \boldsymbol{\theta}^{mm})$, with their outputs concatenated and fused through $\varphi$ to produce the final logit:

$$f(u, v; \boldsymbol{\theta}) = \varphi([g^{id}(\mathbf{e}_u^{id}, \mathbf{e}_v^{id}; \boldsymbol{\theta}^{id}); g^{mm}(\mathbf{e}_u^{mm}, \mathbf{e}_v^{mm}; \boldsymbol{\theta}^{mm})]) \tag{4}$$

We train using binary cross-entropy loss:

$$L = -\frac{1}{N} \sum_{i=1}^{N} \left( y_i \log \hat{y}_i + (1 - y_i) \log(1 - \hat{y}_i) \right), \tag{5}$$

where $\hat{y}_i = \sigma(f(u, v; \boldsymbol{\theta}))$. By decomposing $\varphi$'s weight matrix into $W^{id}$ and $W^{mm}$, we can express Equation 4 as:

$$f(u, v; \boldsymbol{\theta}) = W^{id} * g^{id}(\mathbf{e}_u^{id}, \mathbf{e}_v^{id}; \boldsymbol{\theta}^{id}) + W^{mm} * g^{mm}(\mathbf{e}_u^{mm}, \mathbf{e}_v^{mm}; \boldsymbol{\theta}^{mm}) + b \tag{6}$$

As shown in Appendix C, the ID and MM branches update nearly independently. However, ID features converge faster with stronger signals, dominating predictions and gradient updates. This creates a feedback loop where MM components receive weakened optimization signals, remaining under-utilized at convergence. Consequently, the ID pathway determines the final logit while the MM pathway stays inadequately optimized, causing the observed convergence inconsistency.

**Mask ratio via modal significance**  As discussed, ID and MM representations carry signals of varying strengths during training. To quantify the influence of different representations within the training objective, we introduce a contribution score $s$. For a given sample, the contribution score is formulated as:

$$s = y * p + (1 - y) * (1 - p), \quad \text{where} \quad p = \sigma(W * g(\mathbf{e}_u, \mathbf{e}_v; \boldsymbol{\theta}) + b/2) \tag{7}$$

The relative contribution ratios $\rho^{id}$ and $\rho^{mm}$ are calculated per mini-batch $\mathcal{B}$:

$$\rho^{id} = \frac{\sum_{x_i \in \mathcal{B}} s_i^{id}}{\sum_{x_i \in \mathcal{B}} s_i^{mm}}, \quad \rho^{mm} = 1/\rho^{id} \tag{8}$$

To prevent abrupt fluctuations, we employ exponential moving average (EMA) smoothing with momentum $\lambda$ when updating these ratios across iterations.

$$\rho_t = \lambda \rho_t + (1 - \lambda)\rho_{t-1} \tag{9}$$

In order to mitigate optimization imbalance, we need to provide adequate optimization opportunities to the non-dominant modality while suppressing the parameter updates of the dominant modality. Therefore, inspired by softmax normalization [56], we define the update ratio $\gamma^{id}$ and $\gamma^{mm}$ as follows:

$$\gamma^{id} = \frac{\exp(\rho^{mm})}{\exp(\rho^{id}) + \exp(\rho^{mm})}, \quad \gamma^{mm} = 1 - \gamma^{id} \tag{10}$$

A higher value of $\gamma$ indicates fewer parameters are frozen and more parameters are updated in the corresponding branch.

**Adaptive Gradient Masking**  To implement modality-specific gradient updates, we utilize the Fisher Information Matrix (FIM) [57] , donated as $\mathbf{F}(\boldsymbol{\theta})$, which allows us to effectively measure the relative importance of model parameters across modalities. Specifically, the Fisher Information Matrix is defined as:

$$\mathbf{F}(\boldsymbol{\theta}) = \mathbb{E}\left[\left(\frac{\partial \log p(\hat{y} \mid x; \boldsymbol{\theta})}{\partial \boldsymbol{\theta}}\right)\left(\frac{\partial \log p(\hat{y} \mid x; \boldsymbol{\theta})}{\partial \boldsymbol{\theta}}\right)^{\top}\right] \tag{11}$$

Following [58], given a batch of data, we estimate the importance of parameters using the diagonal elements of $\mathbf{F}(\boldsymbol{\theta})$. Formally, the Fisher information for the $j$-th parameter is calculated as follows:

$$\mathbf{F}_j(\boldsymbol{\theta}) = \frac{1}{|\mathcal{B}|} \sum_{i=1}^{|\mathcal{B}|} \left(\frac{\partial \log p(\hat{y}_i \mid x_i; \boldsymbol{\theta})}{\partial \boldsymbol{\theta}_j}\right)^2 \tag{12}$$

Subsequently, we normalize these diagonal elements and denote $\pi_j$ as the importance score of the $j$-th parameter. Aggregating the importance scores of all parameters, we obtain a probability distribution $\boldsymbol{\pi}$ over the parameter space:

$$\boldsymbol{\pi} = \{\pi_1, \pi_2, ..., \pi_{|\boldsymbol{\theta}|}\}, \quad \text{where} \quad \pi_j = \frac{F_j(\boldsymbol{\theta})}{\sum_{j=1}^{|\boldsymbol{\theta}|} F_j(\boldsymbol{\theta})} \tag{13}$$

Given the update ratio $\gamma$ of each modality and the parameter-wise probability distribution $\boldsymbol{\pi}$, we employ the non-uniform adaptive sampling [59] to generate the gradient mask $\mathbf{m}(t) \in \{0, 1\}^{|\boldsymbol{\theta}|}$, where 1 indicates the parameter will be updated during backpropagation and 0 means it remains frozen. This sampling method primarily directs our focus toward parameters carrying richer information. Concurrently, probabilistic sampling extends coverage to a broader range of parameters, promoting more thorough exploration and enhancing the model's generalization capability across the entire parameter space.

Finally, the parameter update rule with gradient masking thus becomes:

$$\boldsymbol{\theta}(t + 1) = \boldsymbol{\theta}(t) - \eta * \nabla \mathcal{L}(\boldsymbol{\theta}(t)) * \mathbf{m}(t) \tag{14}$$

## 3.4 Theoretical analysis and AGM*

In this section, we present our theoretical analysis of the Asymmetric Gradient Masking (AGM) approach and its improved variant AGM. We begin by establishing the convergence properties of the original AGM method, then introduce an importance weighting scheme that leads to better convergence guarantees in AGM. The convergence of AGM is complicated by the bias introduced through gradient masking. Theorem 1 formalizes this behavior:

**Theorem 1** (Convergence of AGM). *Suppose the loss function $\mathcal{L}(\cdot)$ is L-smooth and $\nabla l(\boldsymbol{\theta}(t))$ is unbiased, i.e. $\mathbb{E}(\nabla l(\boldsymbol{\theta}(t))) = \nabla \mathcal{L}(\boldsymbol{\theta}(t))$, which is commonly used in non-convex optimization. However, 0/1 mask makes $\nabla \ell(\boldsymbol{\theta}(t)) \odot \mathbf{m}(t)$ biased, i.e., $\mathbb{E}[\nabla \ell(\boldsymbol{\theta}(t)) \odot \mathbf{m}(t)] \neq \nabla \mathcal{L}(\boldsymbol{\theta}(t))$, since $\nabla l(\boldsymbol{\theta}(t))$ and $\mathbf{m}(t)$ are not independent. Under the Mask-Incurred Error assumption, we have the following convergence result for AGM over $T$ steps:*

$$\frac{1}{T} \sum_{t=1}^{T} \mathbb{E}[\|\nabla \mathcal{L}(\boldsymbol{\theta}(t))\|^2] \leq \mathcal{O}\left(\frac{1 + (1+\nu)^2}{(1-\delta^2)(1+\nu)\sqrt{T}}\right), \tag{15}$$

*where $\delta \in (0,1)$ and $\nu \geq 0$ are two constants.*

The key limitation here is the bias in gradient estimates caused by the interaction between the mask $\mathbf{m}(t)$ and the stochastic gradients. This bias manifests in the $(1-\delta^2)$ term in the denominator, which slows down convergence. To address this issue, we propose AGM* which incorporates importance weighting through a modified mask $\hat{\mathbf{m}}(t)$. The weights are defined as:

$$\hat{\mathbf{m}}_j(t) = \begin{cases} \dfrac{1}{\pi_j + c}, & \text{if } \mathbf{m}_j(t) = 1, \\ 0, & \text{otherwise.} \end{cases} \tag{16}$$

where $\pi_j$ represents the probability of the $j$-th parameter being unmasked and $c$ is a small constant for numerical stability. This weighting scheme helps compensate for the bias introduced by the original masking operation. The update rule for AGM* becomes:

$$\boldsymbol{\theta}(t+1) = \boldsymbol{\theta}(t) - \eta \nabla \mathcal{L}(\boldsymbol{\theta}(t)) \odot \hat{\mathbf{m}}(t). \tag{17}$$

The importance weighting in AGM* leads to better theoretical guarantees, as shown in Theorem 2:

**Theorem 2** (Convergence of AGM*). *Under some assumptions for $\nabla \ell(\boldsymbol{\theta}(t)) \odot \mathbf{m}(t)$, we have:*

$$\frac{1}{T} \sum_{t=1}^{T} \mathbb{E}[\|\nabla \mathcal{L}(\boldsymbol{\theta}(t))\|^2] \leq \mathcal{O}\left(\frac{1 + (1+\nu)^2}{(1+\nu)\sqrt{T}}\right), \tag{18}$$

Comparing Theorems 1 and 2, we see that AGM* removes the problematic $(1-\delta^2)$ term from the denominator, leading to faster convergence. This improvement comes from the fact that the importance weights in $\hat{\mathbf{m}}(t)$ help maintain the unbiasedness of the gradient estimates despite the masking operation. The complete proofs and additional technical details can be found in the Appendix.

## 4 Experiments

### 4.1 Setup

**Dataset** We conduct offline experiments on four open-source datasets from diverse recommendation domains. First, we choose the Microlens dataset [60], which features user-item interactions, video introductions, and video cover images. In addition, we adopt three categories from the Amazon dataset–Baby, Sports, and Electronics [61, 62]–which contain user-item interactions, product descriptions, and images. All raw datasets are preprocessed with a 5-core setting on both items and users, as described in [12, 10]. Detailed statistics of the datasets are provided in Appendix A.1.

**Baselines and Evaluation** In our experiments, we conduct two parts of evaluation. We first compare AGM with several recent advanced multimodal recommendation models, including VBPR [13], BM3 [10], FREEDOM [9], AlignRec [12], MGCN [8], LGMRec [26], GUME [25], and MM-Rec [63], to demonstrate its effectiveness. Next, to examine the generalization capability of AGM, we test our framework with diverse backbones, including MLP [64], DCN [65], and Fibinet [66]. To evaluate the performance of all models, we adopt two widely-used classification metrics: AUC (Area Under the ROC Curve) [67] and LogLoss (Logarithmic Loss) [68].

Table 1: Performance comparison of AGM and ID, MM, ID+MM models across different backbone architectures, measured by AUC.

| Model | MLP | | | | DCN | | | | Fibinet | | | |
|---|---|---|---|---|---|---|---|---|---|---|---|---|
| | Baby | Elec. | Sports | Micro. | Baby | Elec. | Sports | Micro. | Baby | Elec. | Sports | Micro. |
| ID | 0.6741 | 0.7215 | 0.7012 | 0.6883 | 0.6696 | 0.7192 | 0.6854 | 0.6825 | 0.6792 | 0.7196 | 0.7043 | 0.6901 |
| MM | 0.6237 | 0.6577 | 0.6295 | 0.6279 | 0.6202 | 0.6527 | 0.6251 | 0.6227 | 0.6154 | 0.6548 | 0.6223 | 0.6245 |
| ID+MM | 0.6719 | 0.7218 | 0.7072 | 0.6904 | 0.6685 | 0.7123 | 0.6827 | 0.6857 | 0.6641 | 0.7269 | 0.7115 | 0.6915 |
| AGM* | **0.6864** | **0.7308** | **0.7145** | **0.6992** | **0.6827** | **0.7256** | **0.7062** | **0.6965** | **0.6832** | **0.7310** | **0.7129** | **0.6973** |

Table 2: Comparison of AUC (higher is better) and LogLoss (lower is better) between AGM and other multimodal recommendation methods. Boldface indicates the best performance, and underlined values indicate the second-best.

| Model | Baby | | Electronics | | Sports | | MicroLens | |
|---|---|---|---|---|---|---|---|---|
| | AUC | Logloss | AUC | Logloss | AUC | Logloss | AUC | Logloss |
| VBPR | 0.6729 | 0.6739 | 0.7158 | 0.6032 | 0.6985 | 0.6533 | 0.6758 | 0.6054 |
| FREEDOM | 0.6802 | 0.6708 | 0.7221 | 0.5973 | 0.7023 | 0.6472 | 0.6772 | 0.5957 |
| BM3 | 0.6715 | 0.6712 | 0.7119 | 0.6084 | 0.6932 | 0.6515 | 0.6705 | 0.6021 |
| AlignRec | 0.6832 | 0.6681 | 0.7274 | 0.5988 | 0.7101 | 0.6438 | 0.6869 | 0.5906 |
| MGCN | 0.6810 | 0.6695 | 0.7239 | 0.5994 | 0.7085 | 0.6417 | 0.6851 | 0.5881 |
| LGMRec | 0.6823 | 0.6687 | 0.7247 | 0.6012 | 0.7009 | 0.6480 | 0.6778 | 0.5935 |
| GUME | 0.6834 | 0.6679 | 0.7270 | 0.5991 | 0.7119 | 0.6399 | 0.6968 | 0.5876 |
| MM-Rec | 0.6643 | 0.6691 | 0.7136 | 0.5987 | 0.6703 | 0.6415 | 0.6735 | 0.5885 |
| AGM | 0.6852 | 0.6683 | 0.7285 | 0.5974 | 0.7126 | 0.6405 | 0.6974 | 0.5856 |
| AGM* | **0.6864** | **0.6656** | **0.7310** | **0.5969** | **0.7145** | **0.6391** | **0.6992** | **0.5841** |
| $\Delta_{AGM*-AGM}$ | +0.0012 | -0.0027 | +0.0025 | -0.0005 | +0.0019 | -0.0014 | +0.0018 | -0.0015 |

## 4.2 Performance Comparison

**Compared to different backbones** Table 1 illustrates the AUC performance of four methods (AGM* and other three traditional model frameworks) evaluated on different backbone architectures. More specifically, to analyze the individual and combined effects of different training features and compare their performance to AGM*, we conducted experiments on three traditional model frameworks: (i) ID: A baseline model that utilizes only ID features (e.g., user ID, item ID). (ii) MM: A variant that relies solely on multimodal features (e.g., image, text). (iii) ID+MM: A straightforward combination of ID and multimodal features, without specialized fusion or alignment mechanisms.

The results reveal the following key insights: (i) AGM* consistently achieves the highest AUC across all backbone architectures, demonstrating that AGM* not only achieves superior performance but also maintains robustness and generalizability across various backbones. (ii) Models that rely solely on multimodal features (MM) consistently exhibit the lowest AUC scores across all settings. This suggests that multimodal signals alone are insufficient to capture user preferences, likely due to noise and sparse semantics in text or image modalities.

**Compared to different baselines** Table 2 presents the AUC and LogLoss results of our proposed AGM* and AGM framework in comparison with several state-of-the-art multimodal recommendation baselines across four benchmark datasets. From the experimental results, we derive the following observations: (i) AGM* consistently achieves the best performance on all datasets, although it is based on relatively simple neural network architectures. These gains can be attributed to AGM*'s ability to adaptively modulate feature contributions during training. (ii) Compared with other MLLM-based methods such as AlignRec [12] and GUME [25], AGM* achieves consistently better performance across all datasets. This superiority can be partially attributed to our fine-tuning strategy, which enhances the semantic alignment of multimodal features and ensures better adaptation of the pretrained MLLM to the recommendation domain. (iii) Experimental results prove the effectiveness of our proposed unbiased version AGM*, as it outperforms the biased AGM.

## 4.3 Ablation Study

In this part, we conduct ablation studies to evaluate the contribution of each core component in AGM and AGM*. Specifically, we compare the full models with the following variants: For AGM: (i)

Table 3: Ablation results (AUC) of different modules in AGM.

| Dataset | ID+MM | Random | w/o BM | w/o FM | AGM | w/o BM* | w/o FM* | AGM* |
|---------|-------|--------|--------|--------|-----|---------|---------|------|
| Baby | 0.6719 | 0.6792 | 0.6773 | 0.6815 | **0.6852** | 0.6789 | 0.6837 | **0.6864** |
| Elec. | 0.7218 | 0.7267 | 0.7252 | 0.7271 | **0.7285** | 0.7254 | 0.7281 | **0.7308** |
| Sports | 0.7072 | 0.7103 | 0.7095 | 0.7112 | **0.7126** | 0.7107 | 0.7121 | **0.7145** |
| Micro. | 0.6904 | 0.6949 | 0.6938 | 0.6954 | **0.6974** | 0.6946 | 0.6967 | **0.6992** |

Table 4: The performance of Online A/B Testing at the platform.

| Main | watch-time | app usage | long view | short view |
|------|-----------|-----------|-----------|------------|
| +MM | +0.022% | +0.008% | +0.132% | -0.160% |
| AGM* | +0.175% | +0.124% | +0.678% | -0.235% |

**Random**: When generating the gradient mask $\mathbf{m}(t)$, this variant adopts purely random sampling that ignores parameter importance distributions, instead of the non-uniform adaptive sampling [59]. (ii) **w/o Backbone Masking (-BM)**: This variant disables gradient modulation on the backbone1,2 in Fig. 3. In other words, no additional gradient masking is applied to the backbone network layers; the gradients flow through these two layers in their original form. (iii) **w/o Fusion Masking (-FM)**: This variant omits gradient modulation on the fusion block in Fig. 3, so the gradients are propagated through this block in their original form without adaptive gradient masking. For AGM*, we apply the same ablations: (iv) **w/o Backbone Masking (-BM*)**: Gradient modulation on the backbone1,2 is removed in AGM*. (v) **w/o Fusion Masking (-FM*)**: Gradient modulation on the fusion block is removed in AGM*.

Table 3 presents the experimental results, demonstrating two key observations: (i) The removal of any module leads to a noticeable drop in AUC performance, from which we can conclude that all components make contributions to AGM and AGM*. (ii) Among all the ablation variants, removing the dynamic gradient masking on the backbone1,2 (-BM/-BM*) results in the most significant performance drop. This may be because without proper gradient regulation at this early level, imbalanced learning signals can lead to biased feature extraction from each modality. Consequently, these biases are carried forward and accumulated through the subsequent layers which substantially undermines the downstream fusion process, leading to a more pronounced overall performance loss compared to removing other modules.

## 4.4 Industrial Application

To further assess the real-world effectiveness of our model, we integrate AGM into the industrial recommendation system of a large-scale short video platform that serves hundreds of millions of users. The model is deployed in a 14-day online A/B test to evaluate its performance in a production environment.

We adopt widely-used industry metrics, such as app usage time and watch time, to measure performance. As shown in Table 4, our model achieves substantial improvements over the baseline, further confirming AGM's effectiveness. Notably, the model has now been fully deployed across the platform, actively serving hundreds of millions of users every day.

## 5 Conclusion

In this paper, we tackle the convergence inconsistency problem in joint training of ID-based and MLLM-based representations within large-scale recommendation systems. We propose a two-stage framework that first learns semantically aligned multimodal representations through MLLM fine-tuning and post-alignment with ID features, and then introduces a novel Adaptive Gradient Masking (AGM) strategy to balance optimization across modalities. Our theoretical analysis and extensive empirical results—across both offline benchmarks and real-world A/B testing—demonstrate that the proposed framework effectively mitigates the convergence gap, stabilizes training, and significantly boosts recommendation performance. These findings highlight the importance of

coordinated optimization in multimodal recommendation and pave the way for more robust integration of pretrained models into industrial systems.

## Acknowledgement

This work was partially supported by the Yonghua Foundation.

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

# Supplemental Material: Adaptive Gradient Masking for Balancing ID and MLLM-based Representations in Recommendation

## A  Experimental details

### A.1  Dataset

The detailed descriptions of the datasets used in the main text are as follows:

- **Amazon Baby**: This dataset consists of user-generated reviews on baby-related products sold on Amazon, such as bottles, diapers, and infant toys. It typically includes product names, review texts, and star ratings, making it a valuable resource for research in recommendation systems.

- **Amazon Sports**: This dataset includes Amazon user reviews pertaining to sports and outdoor products, including equipment, camping gear, and fitness devices. Alongside textual reviews and ratings, the dataset provides insights into consumer preferences and opinions, supporting various applications in recommendation modeling.

- **Amazon Electronics**: This dataset is derived from Amazon reviews on electronic products, covering items such as mobile phones, cameras, laptops, and home appliances. In addition to review content and ratings, it contains product metadata, facilitating extensive research in product recommendation.

- **Microlens**: This dataset contains user–item interaction records, video introductions, and video cover images. Each video within MicroLens contains multiple modalities, including text descriptions, images, audio, and raw video information. This extensive coverage enables robust benchmarking of both classical and state-of-the-art recommendation systems.

Table 5: Statistical Information of datasets.

| Dataset | Baby | Sports | Electronics | Microlens |
|---|---|---|---|---|
| #User | 19,445 | 35,598 | 192,403 | 25,411 |
| #Item | 7,050 | 18,357 | 63,001 | 20,276 |
| #Interaction | 160,792 | 296,337 | 1,689,188 | 223,263 |
| Sparsity | 99.88% | 99.95% | 99.99% | 99.96% |

### A.2  Baselines

We summarize the key characteristics of the baseline methods used in our comparative evaluation:

- **VBPR** [13]: This model incorporates visual features into matrix factorization by treating them as auxiliary information for user preference, learning with BPR loss.

- **BM3** [10]: This model simplifies the self-supervised multimodal recommendation framework by using a latent representation dropout mechanism instead of graph augmentation to generate contrastive views, enhancing representation learning.

- **FREEDOM** [9]: This model refines ID-based item representations through auxiliary contrastive losses on multimodal data, and leverages graph structures to align different modalities in a unified embedding space.

- **AlignRec** [12]: This model addresses modality misalignment and optimization imbalance by proposing a two-stage training strategy that first pre-trains inter-content alignment, then jointly optimizes with the recommendation task.

- **MGCN** [8]: This model purifies modality features using item behavior data to reduce noise, and models user-modal preference through multi-view graph convolution networks.
- **LGMRec** [26]: This model enhances multimodal recommendation by learning both local and global semantic relations in item-user graphs, effectively capturing fine-grained user preferences.
- **GUME** [25]: This model improves long-tail multimodal recommendation by incorporating user-specific modality preferences and behavior graphs to enhance personalized modeling.
- **MM-Rec** [63]: This model enhances multimodal news recommendation by jointly encoding news text and image ROIs using a visiolinguistic model, and introduces a candidate-aware attention mechanism to identify relevant historical news.

## A.3 Training details

The code and model are available at: **AGM**.

In the fine-tuning phase, we adopted Qwen2vl-2b[51] as the backbone model. The model was fine-tuned for 5 epochs across four distinct datasets, utilizing a batch size of 128 on 4 A100 GPUs.

For AGM, offline evaluations were conducted using TensorFlow 2.15.0 on a single RTX 4090 GPU, selecting Adam as the optimizer. Hyperparameters, including batch size and learning rate, were systematically tuned across candidate sets of $\{256, 512, 1024, 2048\}$ and $\{1e-3, 1e-4, 1e-5\}$, respectively. The best model was selected based on the minimum validation loss, and early stopping was applied with a patience of 5 to prevent over-fitting.

## B  In-depth analysis

### B.1  Evaluation of multimodal representations

To assess the quality of the multimodal representations generated by AGM, we conduct zero-shot recommendation experiments on the Amazon dataset following the protocol of AlignRec [12]. For each user, the last interacted item is regarded as the target item, and the rest form the historical sequence. We average the multimodal features of historical items to construct the user representation, then compute its similarity with the candidate items' features to evaluate if the target item ranks within top-K. We compare the following methods: (i) Amazon, which uses separately trained visual (CNN) and textual (Transformer) encoders; (ii) MLLM, which directly uses the frozen MLLM outputs; (iii) w/o $\mathcal{L}_{align}$, a variant of AGM that disables the feature alignment loss; and (iv) Ours (AGM), which includes all proposed components. We report Recall@20 and Recall@50 in Table 6. The results reveal three key observations: (i) AGM significantly outperforms traditional feature extractors and raw MLLM features, indicating the benefit of task-specific representation refinement. (ii) Removing the alignment loss leads to noticeable performance drops, highlighting its importance in guiding effective feature selection and fusion. (iii) AGM achieves consistent improvements across all categories, demonstrating its robust capability in modeling multimodal user-item relationships in a zero-shot scenario.

Table 6: Evaluation of multimodal representations

| Generation Methods | Baby | | Elec. | | Sports | |
|---|---|---|---|---|---|---|
| | R@20 | R@50 | R@20 | R@50 | R@20 | R@50 |
| Amazon | 0.0052 | 0.0150 | 0.0093 | 0.0135 | 0.0040 | 0.0072 |
| MLLM | 0.0140 | 0.0345 | 0.0202 | 0.0364 | 0.0053 | 0.0089 |
| w/o $\mathcal{L}_{align}$ | 0.0225 | 0.0426 | 0.0251 | 0.0417 | 0.0120 | 0.0159 |
| Ours | **0.0276** | **0.0509** | **0.0293** | **0.0478** | **0.0175** | **0.0206** |

### B.2  Convergence analysis

To better illustrate the convergence performance of different methods, we plot the AUC values across training epochs for AGM, AGM*, and the combined model (ID-MM) on Amazon Baby, Electronics, Sports and Microlens. As shown in Fig.4, both AGM and AGM* demonstrate increasing

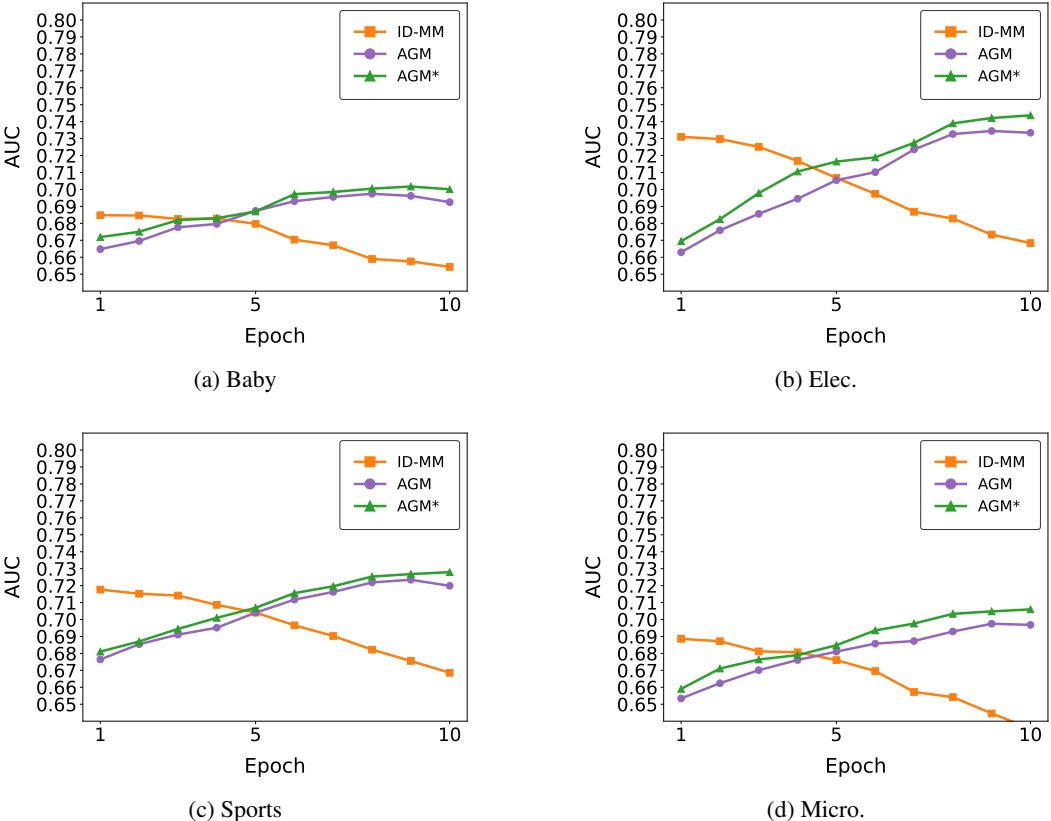

Figure 4: Validation-set AUC comparison of the combined model (ID-MM), AGM, and AGM* across four datasets

AUC throughout training epochs, indicating effective joint learning. In contrast, ID-MM exhibits a downward trend in AUC as training progresses. This degradation can be attributed to the ID features dominating the training process of ID-MM, especially in the absence of any modulation mechanism on gradients during the backpropagation phase. Consequently, as the model overfits the ID space over time, the performance of ID-MM degrades. This dominance suppresses the contribution of multimodal features, resulting in suboptimal representations and overall performance decline. In contrast, our AGM and AGM* introduce Gradient Modulation, which dynamically balances the gradient flow between the ID and multimodal branches, preventing ID features from overwhelming MM features, allowing both to contribute meaningfully to the final prediction.

In addition, AGM* shows better overall performance compared to AGM, especially in larger datasets (e.g., Amazon Electronics). Specifically, by compensating for the gradient masking bias introduced by the original masking operation in the AGM, AGM* improves the gradient update process, promoting faster convergence, and ultimately improving the final AUC of the model.

## C  Convergence inconsistency analysis of ID-MM combine model

In the joint learning of ID and multimodal representations for recommendation, we observe an optimization imbalance phenomenon, where one representation dominates the learning process, causing the other to be under-optimized. We introduce the analysis of the optimization imbalance phenomenon for the model with concatenation as fusion method. In our recommendation model, the logits output is formulated as:

$$\varphi(x_i) = W[f(\mathbf{e}^{id}; \boldsymbol{\theta}^{id}); g(\mathbf{e}^{mm}; \boldsymbol{\theta}^{mm})] + b \tag{19}$$

To observe the optimization process of each component individually, $W$ can be represented as the combination of two matrix: $W^{id}$ and $W^{mm}$. The Equation 19 can be rewritten as:

$$\varphi(x_i) = W^{id} * f(\mathbf{e}^{id}; \boldsymbol{\theta}^{id}) + W^{mm} * g(\mathbf{e}^{mm}; \boldsymbol{\theta}^{mm}) + b \tag{20}$$

The model is trained using binary cross-entropy loss:

$$L = -\frac{1}{N} \sum_{i=1}^{N} \left( y_i \log \hat{y}_i + (1 - y_i) \log(1 - \hat{y}_i) \right), \tag{21}$$

where the predicted probability is $\hat{y} = \sigma(\varphi(x_i))$. The gradient of the loss with respect to the logit is:

$$\frac{\partial L}{\partial \varphi(x_i)} = \sigma(\varphi(x_i)) - y_i. \tag{22}$$

Applying the chain rule, the gradients for the ID weights and backbone parameters of the ID representation are as follows:

$$\frac{\partial L}{\partial W^{id}} = \frac{1}{N} \sum_{i=1}^{N} \frac{\partial L}{\partial \varphi(x_i)} * f(\mathbf{e}^{id}; \boldsymbol{\theta}^{id}) \tag{23}$$

$$\frac{\partial L}{\partial \boldsymbol{\theta}^{id}} = \frac{1}{N} \sum_{i=1}^{N} \frac{\partial L}{\partial \varphi(x_i)} W^{id} \frac{\partial f(\mathbf{e}^{id}; \boldsymbol{\theta}^{id})}{\partial \boldsymbol{\theta}^{id}} \tag{24}$$

According to Equations 23 and 24, the optimization of $W^{id}$ and $f(\cdot)$ is nearly independent of the optimization of multimodal parameters $W^{mm}$ and $g(\cdot)$, except for the term associated with the training loss. As a result, the backbone have limited ability to make adjustments based on feedback from one another. The analysis of the feed-forward and back-propagation stages reveals that both the model predictions and gradients are governed by the sum of ID and multimodal (MM) components. Since ID features converge faster and contain stronger discriminative information, they dominate the model prediction $\varphi(x_i)$ and gradient $\frac{\partial L}{\partial \varphi(x_i)}$ through $W^{id} \cdot f(\mathbf{e}^{id}; \boldsymbol{\theta}^{id})$. Even if the MM representations remain under-optimized and produce erroneous outputs during training, the more informative ID components can still "correct" these errors through summation, thereby influencing both the feed-forward and back-propagation processes. Consequently, according to Eq. (9) and Eq. (11), the MM, which has relatively lower confidence in the correct category, receives limited optimization, leading to its under-utilization. Based on this analysis, ID features play a dominant role in the optimization process. As the model approaches convergence, MM components may still require further training to compensate for their under-optimized features.

## D    Convergence analysis of AGM

In this section, we give a detailed proof of Theorem 1. Recall update step of stochastic gradient descent (SGD) for AGM is:

$$\boldsymbol{\theta}(t + 1) = \boldsymbol{\theta}(t) - \eta \cdot \nabla \ell(\boldsymbol{\theta}(t)) \odot \mathbf{m}(t), \tag{25}$$

where $\nabla \ell(\boldsymbol{\theta}(t))$ is the stochastic gradient of $\nabla \mathcal{L}(\boldsymbol{\theta}(t))$ and $\eta > 0$ is the learning rate, and $\mathbf{m}(t)$ is a binary mask vector. To analyze the convergence of AGM, we have the three following common assumptions for $\mathcal{L}(\cdot)$.

**Assumption 1** (Smoothness). *The loss function $\mathcal{L}$ is $L$-smooth, which is common for non-convex optimization. That is, for any $\boldsymbol{\theta}, \boldsymbol{\theta}'$, we have:*

$$\mathcal{L}(\boldsymbol{\theta}) - \mathcal{L}(\boldsymbol{\theta}') \leq \langle \nabla \mathcal{L}(\boldsymbol{\theta}'), \boldsymbol{\theta} - \boldsymbol{\theta}' \rangle + \frac{L}{2} \|\boldsymbol{\theta} - \boldsymbol{\theta}'\|^2. \tag{26}$$

**Assumption 2** (Bounded Variance). *We assume that the stochastic gradient $\nabla \ell(\boldsymbol{\theta}) \odot \mathbf{m}(t)$ is biased and its variance is bounded. That is, for any $\boldsymbol{\theta}(t)$ and $\mathbf{m}(t)$, we have*

$$\mathbb{E} \left[ \nabla \ell(\boldsymbol{\theta}(t)) \odot \mathbf{m}(t) \right] = \nabla \mathcal{L}(\boldsymbol{\theta}(t)) + b(\boldsymbol{\theta}(t)), \tag{27}$$

*and*

$$\mathbb{E} \left[ \|\nabla \ell(\boldsymbol{\theta}(t)) \odot \mathbf{m}(t) - \mathbb{E} \left[ \nabla \ell(\boldsymbol{\theta}(t)) \odot \mathbf{m}(t) \right] \|\right]^2 \leq \nu \|\nabla \mathcal{L}(\boldsymbol{\theta}(t)) + b(\boldsymbol{\theta}(t))\|^2 + \sigma^2, \tag{28}$$

*where $\sigma^2 \geq 0$ and $\nu \geq 0$ are two constants.*

**Assumption 3** (Mask-Incurred Error). *For any $\boldsymbol{\theta}(t)$ and $\mathbf{m}(t)$, we have*

$$\|\mathbb{E} \left[ \nabla \ell(\boldsymbol{\theta}(t)) \odot \mathbf{m}(t) - \nabla \ell(\boldsymbol{\theta}(t)) \right] \| \leq \delta \|\mathbb{E} \left[ \nabla \ell(\boldsymbol{\theta}(t)) \right] \|, \tag{29}$$

*where the constant $\delta \in [0, 1]$.*

**Proof of Theorem 1.** *As Theorem 1 claims:*

$$\frac{1}{T}\sum_{t=1}^{T}\|\nabla\mathcal{L}(\boldsymbol{\theta}(t))\|^2 \le \mathcal{O}\left(\frac{1+(1+\nu)^2}{(1-\delta^2)(1+\nu)\sqrt{T}}\right), \tag{30}$$

By Assumption 1, and let $h(t) := \nabla\ell(\boldsymbol{\theta}(t))\odot\mathbf{m}(t)$, we have

$$\mathcal{L}(\boldsymbol{\theta}(t+1)) - \mathcal{L}(\boldsymbol{\theta}(t)) \le \langle\nabla\mathcal{L}(\boldsymbol{\theta}(t)), \boldsymbol{\theta}(t+1) - \boldsymbol{\theta}(t)\rangle + \frac{L}{2}\|\boldsymbol{\theta}(t+1) - \boldsymbol{\theta}(t)\|^2$$

$$= -\eta\langle\nabla\mathcal{L}(\boldsymbol{\theta}(t)), h(t)\rangle + \frac{\eta^2 L}{2}\|h(t)\|^2, \quad \text{(using eq. 25)} \tag{31}$$

Taking expectation over both sides of 31 and by using Assumption 2:

$$\mathbb{E}[\mathcal{L}(\boldsymbol{\theta}(t+1)) - \mathcal{L}(\boldsymbol{\theta}(t))] \le -\eta\langle\nabla\mathcal{L}(\boldsymbol{\theta}(t)), \nabla\mathcal{L}(\boldsymbol{\theta}(t)) + b(\boldsymbol{\theta}(t))\rangle + \frac{\eta^2 L}{2}\mathbb{E}\left[\|h(t)\|^2\right]$$

$$= -\eta\langle\nabla\mathcal{L}(\boldsymbol{\theta}(t)), \nabla\mathcal{L}(\boldsymbol{\theta}(t)) + b(\boldsymbol{\theta}(t))\rangle + \frac{\eta^2 L}{2}\left(\mathbb{E}[\|h(t) - \mathbb{E}[h(t)]\|^2] + \mathbb{E}[\|\mathbb{E}[h(t)]\|^2]\right)$$

$$\le -\eta\langle\nabla\mathcal{L}(\boldsymbol{\theta}(t)), \nabla\mathcal{L}(\boldsymbol{\theta}(t)) + b(\boldsymbol{\theta}(t))\rangle + \frac{\eta^2 L}{2}\left((1+\nu)\|\nabla\mathcal{L}(\boldsymbol{\theta}(t)) + b(\boldsymbol{\theta}(t))\|^2 + \sigma^2\right)$$

$$\le -\eta\langle\nabla\mathcal{L}(\boldsymbol{\theta}(t)), \nabla\mathcal{L}(\boldsymbol{\theta}(t)) + b(\boldsymbol{\theta}(t))\rangle + \frac{\eta}{2}\|\nabla\mathcal{L}(\boldsymbol{\theta}(t)) + b(\boldsymbol{\theta}(t))\|^2 + \frac{\eta^2 L\sigma^2}{2}, \tag{32}$$

where the last inequality is due to $\eta \le \frac{1}{(1+\nu)L}$.

Since $-\langle a, b\rangle + \frac{\|b\|^2}{2} = \frac{\|a-b\|^2}{2} - \frac{\|a\|^2}{2}$, then 32 implies that

$$\mathbb{E}[\mathcal{L}(\boldsymbol{\theta}(t+1))] - \mathcal{L}(\boldsymbol{\theta}(t)) \le -\eta\langle\nabla\mathcal{L}(\boldsymbol{\theta}(t)), \nabla\mathcal{L}(\boldsymbol{\theta}(t)) + b(\boldsymbol{\theta}(t))\rangle + \frac{\eta^2 L}{2}\mathbb{E}\left[\|h(t)\|^2\right]$$

$$\le \frac{\eta}{2}\|b(\boldsymbol{\theta}(t))\|^2 - \frac{\eta}{2}\|\nabla\mathcal{L}(\boldsymbol{\theta}(t))\|^2 + \frac{\eta^2 L\sigma^2}{2}. \tag{33}$$

Next, by 27 in Assumption 2 and Assumption 3, we know

$$\|b(\boldsymbol{\theta}(t))\| = \|\mathbb{E}[\nabla\ell(\boldsymbol{\theta}(t))\odot\mathbf{m}(t)] - \nabla\ell(\boldsymbol{\theta}(t))\| \le \delta\|\mathbb{E}[\nabla\ell(\boldsymbol{\theta}(t))]\| = \delta\|\nabla\mathcal{L}(\boldsymbol{\theta}(t))\|. \tag{34}$$

Therefore, by (33) and (34) we have

$$\mathbb{E}[\mathcal{L}(\boldsymbol{\theta}(t+1)) - \mathcal{L}(\boldsymbol{\theta}(t))] \le -\frac{\eta(1-\delta^2)}{2}\|\nabla\mathcal{L}(\boldsymbol{\theta}(t))\|^2 + \frac{\eta^2 L\sigma^2}{2}. \tag{35}$$

which implies

$$\|\nabla\mathcal{L}(\boldsymbol{\theta}(t))\|^2 \le \frac{2\mathbb{E}[\mathcal{L}(\boldsymbol{\theta}(t)) - \mathcal{L}(\boldsymbol{\theta}(t+1))]}{\eta(1-\delta^2)} + \frac{\eta L\sigma^2}{1-\delta^2}. \tag{36}$$

By summing up for $t = 1, \ldots, T$, we have

$$\frac{1}{T}\sum_{t=1}^{T}\|\nabla\mathcal{L}(\boldsymbol{\theta}(t))\|^2 \le \frac{2\mathbb{E}[\mathcal{L}(\boldsymbol{\theta}(1)) - \mathcal{L}(\boldsymbol{\theta}(T+1))]}{T\eta(1-\delta^2)} + \frac{\eta L\sigma^2}{1-\delta^2}$$

$$\le \frac{2\mathcal{L}(\boldsymbol{\theta}(1))}{T\eta(1-\delta^2)} + \frac{\eta L\sigma^2}{1-\delta^2}. \tag{37}$$

By setting $\eta = \frac{1}{(1+\nu)L\sqrt{T}}$, we get

$$\frac{1}{T}\sum_{t=1}^{T}\|\nabla\mathcal{L}(\boldsymbol{\theta}(t))\|^2 \le \frac{2(1+\nu)L\mathcal{L}(\boldsymbol{\theta}(1))}{\sqrt{T}(1-\delta^2)} + \frac{\sigma^2}{(1+\nu)\sqrt{T}(1-\delta^2)} = \mathcal{O}\left(\frac{1+(1+\nu)^2}{(1-\delta^2)(1+\nu)\sqrt{T}}\right). \tag{38}$$

# E  Convergence analysis of AGM*

For unbiased stochastic gradient, recall that the update step of SGD is

$$\boldsymbol{\theta}(t+1) = \boldsymbol{\theta}(t) - \eta \nabla \ell(\boldsymbol{\theta}(t)) \odot \hat{\mathbf{m}}(t), \tag{39}$$

where $\nabla \ell(\boldsymbol{\theta}(t))$ is the stochastic version of the gradient of loss function $\nabla \mathcal{L}(\boldsymbol{\theta}(t))$ at $\boldsymbol{\theta}(t)$, and $\eta > 0$ is the learning rate.

The element of $\hat{\mathbf{m}}(t)$ is given by

$$\hat{\mathbf{m}}_j(t) = \begin{cases} \dfrac{1}{\pi_j(t)}, & \text{if } \mathbf{m}_j(t) = 1, \\ 0, & \text{otherwise.} \end{cases} \tag{40}$$

Suppose that stochastic gradient $\nabla \ell(\boldsymbol{\theta}(t))$ is unbiased, i.e., $\mathbb{E}[\nabla \ell(\boldsymbol{\theta}(t))] = \nabla \mathcal{L}(\boldsymbol{\theta}(t))$. Then we have

$$\begin{aligned} \mathbb{E}[\nabla \ell(\boldsymbol{\theta}(t)) \odot \hat{\mathbf{m}}(t)] &= \mathbb{E}[\nabla \ell(\boldsymbol{\theta}(t)) \odot \boldsymbol{\pi}(t)^{-1} \odot \mathbf{m}(t)] \\ &= \mathbb{E}_{\mathbf{m}(t)} \left[ \nabla \ell(\boldsymbol{\theta}(t)) \odot \boldsymbol{\pi}(t)^{-1} \odot \mathbf{m}(t) \mid \nabla \ell(\boldsymbol{\theta}(t)) \right] \\ &= \mathbb{E}[\nabla \ell(\boldsymbol{\theta}(t)) \odot \boldsymbol{\pi}(t)^{-1} \odot \mathbb{E}_{\mathbf{m}(t)}[\mathbf{m}(t) \mid \nabla \ell(\boldsymbol{\theta}(t))]] \\ &= \mathbb{E}[\nabla \ell(\boldsymbol{\theta}(t))] \\ &= \nabla \mathcal{L}(\boldsymbol{\theta}(t)), \end{aligned} \tag{41}$$

indicating that $\nabla \ell(\boldsymbol{\theta}(t)) \odot \hat{\mathbf{m}}(t)$ is also unbiased.

**Assumption 4** (Bounded Variance). *We assume that the stochastic gradient $\nabla \ell(\boldsymbol{\theta}) \odot \hat{\mathbf{m}}(t)$ is unbiased and its variance is bounded. That is, for any $\boldsymbol{\theta}(t)$ and $\hat{\mathbf{m}}(t)$, we have*

$$\mathbb{E}\left[\nabla \ell(\boldsymbol{\theta}(t)) \odot \hat{\mathbf{m}}(t)\right] = \nabla \mathcal{L}(\boldsymbol{\theta}(t)), \tag{42}$$

*and*

$$\mathbb{E}\left[\|\nabla \ell(\boldsymbol{\theta}(t)) \odot \hat{\mathbf{m}}(t) - \mathbb{E}\left[\nabla \ell(\boldsymbol{\theta}(t)) \odot \hat{\mathbf{m}}(t)\right]\|\right]^2 \le \nu \|\nabla \mathcal{L}(\boldsymbol{\theta}(t))\|^2 + \sigma^2, \tag{43}$$

*where $\sigma^2 \ge 0$ and $\nu \ge 0$ are two constants.*

**Proof of Theorem 2.**  As Theorem 2 claims: under assumptions 1, 4, we have:

$$\frac{1}{T} \sum_{t=1}^{T} \|\nabla \mathcal{L}(\boldsymbol{\theta}(t))\|^2 \le \mathcal{O}\left(\frac{1 + (1+\nu)^2}{(1+\nu)\sqrt{T}}\right), \tag{44}$$

where the learning rate is set as $\eta = \frac{1}{(1+\nu)L\sqrt{T}}$.

By Assumption 1, and we set $\hat{h}(t) := \nabla \ell(\boldsymbol{\theta}(t)) \odot \hat{\mathbf{m}}(t)$, then we have

$$\begin{aligned} \mathcal{L}(\boldsymbol{\theta}(t+1)) - \mathcal{L}(\boldsymbol{\theta}(t)) &\le \langle \nabla \mathcal{L}(\boldsymbol{\theta}(t)), \boldsymbol{\theta}(t+1) - \boldsymbol{\theta}(t) \rangle + \frac{L}{2} \|\boldsymbol{\theta}(t+1) - \boldsymbol{\theta}(t)\|^2 \\ &= -\eta \left\langle \nabla \mathcal{L}(\boldsymbol{\theta}(t)), \hat{h}(t) \right\rangle + \frac{\eta^2 L}{2} \left\|\hat{h}(t)\right\|^2, \end{aligned} \tag{45}$$

Taking expectation over both sides of (45) and by using Assumption 4, we have

$$\begin{aligned} \mathbb{E}[\mathcal{L}(\boldsymbol{\theta}(t+1))] - \mathcal{L}(\boldsymbol{\theta}(t)) &\le -\eta \|\nabla \mathcal{L}(\boldsymbol{\theta}(t))\|^2 + \frac{\eta^2 L}{2} \mathbb{E}[\|\hat{h}(t)\|^2] \\ &= -\eta \|\nabla \mathcal{L}(\boldsymbol{\theta}(t))\|^2 + \frac{\eta^2 L}{2} \left( \mathbb{E}\left[\|\hat{h}(t) - \nabla \mathcal{L}(\boldsymbol{\theta}(t))\|^2\right] + \|\nabla \mathcal{L}(\boldsymbol{\theta}(t))\|^2 \right) \\ &\le -\eta \|\nabla \mathcal{L}(\boldsymbol{\theta}(t))\|^2 + \frac{\eta^2 L}{2} \left( (1+\nu)\|\nabla \mathcal{L}(\boldsymbol{\theta}(t))\|^2 + \sigma^2 \right) \\ &\le -\eta \|\nabla \mathcal{L}(\boldsymbol{\theta}(t))\|^2 + \frac{\eta}{2} \|\nabla \mathcal{L}(\boldsymbol{\theta}(t))\|^2 + \frac{\eta^2 L \sigma^2}{2}, \end{aligned} \tag{46}$$

where the last inequality is due to $\eta \leq \frac{1}{(1+\nu)L}$. Therefore, we have

$$\|\nabla\mathcal{L}(\boldsymbol{\theta}(t))\|^2 \leq \frac{2\mathbb{E}[\mathcal{L}(\boldsymbol{\theta}(t)) - \mathcal{L}(\boldsymbol{\theta}(t+1))]}{\eta} + \eta L \sigma^2. \tag{47}$$

By summing up for $t = 1, \ldots, T$, we have

$$\frac{1}{T}\sum_{t=1}^{T}\|\nabla\mathcal{L}(\boldsymbol{\theta}(t))\|^2 \leq \frac{2\mathbb{E}[\mathcal{L}(\boldsymbol{\theta}(1)) - \mathcal{L}(\boldsymbol{\theta}(T+1))]}{T\eta} + \eta L \sigma^2$$

$$\leq \frac{2\mathcal{L}(\boldsymbol{\theta}(1))}{T\eta} + \eta L \sigma^2. \tag{48}$$

Since $\eta = \frac{1}{(1+\nu)L\sqrt{T}}$, we get

$$\frac{1}{T}\sum_{t=1}^{T}\|\nabla\mathcal{L}(\boldsymbol{\theta}(t))\|^2 \leq \frac{2(1+\nu)L\mathcal{L}(\boldsymbol{\theta}(1))}{\sqrt{T}} + \frac{\sigma^2}{(1+\nu)\sqrt{T}}. \tag{49}$$

## F Limitation

While our proposed AGM framework demonstrates promising results, several limitations remain. First, the framework's performance may depend on the quality and diversity of the multimodal data available during training. Second, the current experiments focus on recommendation tasks, and extending the approach to other multimodal applications may require further adaptation.

## G Broader Impacts

Our work on Adaptive Gradient Masking for recommendation systems presents several important societal implications. The improved ability to handle multimodal content could significantly enhance recommendation quality, particularly for niche and cold-start items, potentially benefiting both users through more relevant suggestions and content creators through better exposure. The framework's ability to balance different feature types may also lead to more diverse recommendations, mitigating some common filter bubble effects.

