# OpenReview forum: "Adaptive Gradient Masking for Balancing ID and MLLM-based Representations in Recommendation"
_NeurIPS.cc/2025/Conference — NeurIPS 2025 poster_

### Official Review · Reviewer_fBa2 · 2025-06-30

**Clarity:** 2
**Significance:** 3
**Originality:** 3
**Rating:** 4
**Confidence:** 3

**Summary:**

The semantic distribution gap between ID features and multimodal representations leads to convergence inconsistency during joint training: the ID branch converges quickly, whereas the multimodal branch requires more training rounds, which limits overall performance. To tackle this issue, the paper proposes a two-stage framework consisting of multimodal representation learning and joint training optimization. Specifically, it first fine-tunes a multimodal large language model (MLLM) to generate unified user and item representations, introducing collaborative signals via post-alignment of user ID embeddings to mitigate semantic discrepancies. Then, it introduces an Adaptive Gradient Masking (AGM) training strategy that dynamically adjusts parameter updates between the ID and MLLM branches. AGM estimates the contribution of each representation via mutual information and applies non-uniform gradient masks at the sub-network level to balance optimization.

**Questions:**

Please see the Weaknesses section for detailed questions and suggestions.

**Ethical Concerns:**

["NO or VERY MINOR ethics concerns only"]

**Final Justification:**

After considering the authors' rebuttal, I find that most of my concerns have been satisfactorily addressed. Specifically: I still have some reservations regarding the semantic consistency between $e_u^{mm}$ and $e_v^{mm}$. While the authors argue that $e_u^{mm}$ is derived from historical interactions involving $e_v^{mm}$, and that the alignment loss encourages compatibility, there remains no explicit modeling component (e.g., joint interaction, dot product, or similarity constraint) that ensures both embeddings reside in the same latent space. This may affect downstream interaction modeling, though empirical results do suggest the method works in practice.


  Given that the main methodological concerns have been addressed and the remaining issue is relatively minor and partially mitigated by empirical evidence, I have increased my score from a 3 to a 4. I believe the paper makes a meaningful contribution and could be improved further with minor clarifications and refinements.

**Limitations:**

Yes

**Quality:**

2

**Strengths And Weaknesses:**

## Strengths

* The proposed method effectively addresses the convergence inconsistency between ID and MLLM representations in large-scale recommender systems, leading to improved recommendation performance.
* The paper introduces Adaptive Gradient Masking (AGM) and its theoretically motivated unbiased variant, AGM\*.
* Extensive offline experiments and online A/B testing validate the effectiveness of the proposed approach.

## Weaknesses

* In Section 3.2.1, the authors only describe aligning text-image embeddings within the multimodal item embedding space. Why is there no alignment between the item multimodal embedding and the item ID embedding?
* Additionally, the backbone function $g()$ operates on $e_u$ and $e_i$ for downstream tasks. How does the UEG module ensure that the learned $e_u^{mm}$ and $e_v^{mm}$ reside in the same space? The training tasks for UEG in Section 3.2.2 rely on aligning user multimodal embeddings and ID embeddings, but this does not guarantee that the final $e_u^{mm}$ and $e_v^{mm}$ are in the same space.
* Why does Equation (3) use the L2 norm as the alignment loss? Have the authors considered alternative distance metrics?
* The description in Lines 213–219 is rather vague. For example, “non-uniform adaptive sampling” is not properly introduced. The phrase “This sampling method primarily directs our focus toward parameters carrying richer information” is unclear — is the goal to update these parameters or to preserve them? Finally, what does “probabilistic sampling extends coverage to a broader range of parameters” specifically refer to? Why does this provide broader coverage compared to uniform sampling?
* Could the authors clarify how the range of $\eta$ in Line 661 is determined and why $\eta$ is assigned such a value in Line 667?
* The notation system in Section 3 is confusing:
  * In the problem definition in Section 3.1, the dataset $\mathcal{D}$ uses $u$ and $v$ to indicate clicks $y$, but also uses subscripts $(u_i, v_i, y_i)$, which can lead to inconsistencies if the same user/item appears with different subscripts.
  * In subsequent descriptions, the notation mixes indexed and non-indexed forms for users/items. For instance, in the formula in Line 185, $\hat{y}_i$ is used together with unindexed $u$ and $v$.
  * The variable $x_i$ in Equation (8) is not clearly introduced in context.
  * …
* There are minor writing issues throughout the paper:
  * For example, the abbreviation “MM” in Line 4 should be defined as “multimodal (MM)” upon first use.
  * There are inconsistent capitalization issues, such as in Line 50.
  * …

---

> ### Author Rebuttal · Authors · 2025-07-30
>
> Dear reviewer fBa2,
>
> We sincerely thank you for your careful reading and thoughtful feedback. We truly appreciate the time and effort you dedicated to evaluating our work. Below, we address each of your concerns in detail and provide clarifications where necessary.
>
> **Q1: No alignment between item MM and ID represetations**
>
> User ID and MM representations are both derived from users’ historical behavior sequences, and thus implicitly encode similar semantics — namely, the user’s preferences. Aligning them helps bridge sparse collaborative signals with rich content-derived features, especially in cold-start scenarios.
>
> In contrast, item multimodal features represent "intrinsic, static content semantics", while item ID embeddings are learned via collaborative signals and are continually updated based on user interactions. These two representations capture fundamentally different information: the former reflects what the item is, the latter who it appeals to. Aligning them may dilute the semantic integrity of content features. Therefore, we align user ID and MM but deliberately keep item ID and MM disentangled.
>
>
> **Q2: Latent Space Between $e^{mm}_u$ and $e^{mm}_v$**
>
> In our model,   $e^{mm}_u$ is generated from historical behavior sequences consist of $e^{mm}_v$ , which ensures that they are naturally embedded in a shared latent space compatible for downstream interaction modeling. While the UEG module includes an alignment loss to guide $e^{mm}_u$ toward the ID-based user embedding space, we note that the alignment loss is assigned a small weight in the total objective. This design choice ensures that the core semantic structure of $e^{mm}_u$ remains largely intact and does not significantly deviate from the original MM space.
>
> Furthermore, we provide evidence in Table 6: when we remove the alignment loss, the quality of MM representations drops significantly in the zero-shot recommendation task, which is largely determined by the semantic compatibility between  $e^{mm}_u$ and $e^{mm}_v$. This degradation suggests that adding alignment process between $e^{mm}_u$ and $e^{id}_u$ does not significantly distort the originally consistent semantic space of $e^{mm}_u$ and $e^{mm}_v$, otherwise adding this alignment process would only result in performance loss in MM representation quality in the zero-shot recommendation task. Even if the alignment might slightly influence the original semantic space of $e^{mm}_u$, the benefit of better aligning user representations outweighs any potential disruption, leading to improved recommendation quality.
>
> **Q3: Choice of L2 Norm for Alignment Loss in Equation (3)**
>
> To be honest, we chose the L2 norm as the alignment loss just because it provides a simple, smooth, and well-behaved metric that equally penalizes deviations across all embedding dimensions. This decision aligns with many prior works in embedding alignment, such as AlignRec [1].
>
> **Q4: Unclear description in Lines 213–219**
>
> We apologize for the confusion and clarify here. The “non-uniform adaptive sampling” refers to an importance-based random selection of parameters during training. Concretely, we bias the sampling toward those with higher scores in distribution $\pi$ when we choose which parameters to update during the backforward propagation. In other words, parameters carrying more information are updated more frequently. Therefore, The phrase “directs our focus toward parameters carrying richer information” means that we tend to update these important parameters more often (instead of preserve them).
>
> The second phrase, “probabilistic sampling extends coverage to a broader range of parameters” confused you because the uniform sampling in this occasion is not clear. We apologize for the confusion here. Actully, AGM is fundamentally designed to prioritize the update of more important parameters, and in this context, our “probabilistic sampling” is intended as an improvement over traditional top-k masking strategies commonly used in importance-based sparse updates. Therefore, the sentence“probabilistic sampling extends coverage to a broader range of parameters” means compared to top-k methods, our sampling approach achieves broader coverage of the parameter space.
>
>
> **Q5: $\eta$ in Lines 661 and 667**
>
> To ensure the expected decrease in loss, we consider the worst-case alignment where the bias term $ b(\theta) $ is in the same direction as the loss gradient $ \nabla \mathcal{L}(\theta) $, i.e., $ b = k \nabla \mathcal{L} $.
> In this case, the inner product and norm simplify as:
> $$
> \langle \nabla \mathcal{L}, \nabla \mathcal{L} + b \rangle = (1 + k)\||\nabla \mathcal{L}\||^2,\quad \||\nabla \mathcal{L} + b\||^2 = (1 + k)^2 \||\nabla \mathcal{L}\||^2.
> $$
> By substituting into Eq. (32) (second-to-last line) and neglecting $\sigma$, we obtain:
> $$
> -\eta (1 + k) \||\nabla \mathcal{L}\||^2 + \frac{\eta^2 L}{2}(1 + \nu)(1 + k)^2 \||\nabla \mathcal{L}\||^2 \leq 0.
> $$
> Dividing both sides by $ (1 + k)\||\nabla \mathcal{L}\||^2 $, we get:
> $$
> \eta \leq \frac{2}{L(1 + \nu)(1 + k)}.
> $$
> By conservatively taking $ k = 1 $, we obtain the sufficient condition:
> $$
> \eta \leq \frac{1}{L(1 + \nu)}.
> $$
> This justifies the learning rate constraint adopted in the analysis.
>
> The upper bound in Line 661, $ \eta \leq \frac{1}{(1 + \nu)L} $, ensures descent of the expected loss under worst-case bias alignment.
> In Line 667, this setting serves two purposes:
> 1. It satisfies the sufficient condition for convergence established in Line 661.
> 2. It follows the standard choice in stochastic optimization to achieve an $ \mathcal{O}(1/\sqrt{T}) $ convergence rate for the average iterate. This rate matches the optimal bound under noisy gradient conditions, and the $ 1/(1 + \nu) $ scaling further accounts for the additional bias variance introduced in our problem setting.
>
> **Q6: Confusing notation**
>
> We sincerely apologize for the confusion this may have caused. Regarding the specific points you raised:
> (i) The variable $x_i$ in Equation (8) refers to the current training sample, and corresponds to the tuple ($u_i$, $v_i$). We will clarify this in the revised version.
> (ii) The notation alternates between indexed and unindexed forms (e.g., $u_i$, $v_i$ vs. $u$, $v$), which may lead to ambiguity. In the revision, we will ensure that all sample-specific variables are consistently indexed, and clarify where variables represent arbitrary users/items versus specific samples.
>
> We appreciate your attention to these important details and will revise Section 3 accordingly to improvev readability and consistency.
>
> **Q7: Writing issues**
>
> We sincerely thank you for pointing out the writing and formatting issues, including the missing definition of “MM” (Line 4), inconsistent capitalization (Line 50), and other stylistic inconsistencies. We apologize for these oversights. These corrections will be carefully incorporated into the revised version to ensure clarity, consistency, and overall presentation quality.
>
> Finally, we truly appreciate all your valuable feedback. We hope our responses have addressed your concerns, and we would be grateful if you could take these clarifications into account in your overall evaluation. Thank you again for your time and insights — we look forward to any further comments you may have.
>
> [1] AlignRec: Aligning and Training in Multimodal Recommendations

---

> > ### Comment · Reviewer_fBa2 · 2025-08-04
> >
> > Thank you for the detailed response. I appreciate that most of my concerns have been addressed. Regarding Q2, I now understand that $e_u^{mm}$ is derived from the user’s historical interactions with $e_v^{mm}$, which implicitly encourages compatibility between the two modalities. However, I still have some reservations: the model does not seem to include any direct operations—such as dot-product or joint interaction modeling—between $e_u^{mm}$ and $e_v^{mm}$ that would explicitly enforce them to lie in the same semantic space.
> >
> > Therefore, although the use of alignment loss on $e_u^{mm}$ is helpful, it remains unclear whether $e_u^{mm}$ and $e_v^{mm}$ are guaranteed to reside in a shared latent space suitable for downstream interaction modeling.
> >
> > That said, given that you have adequately addressed the majority of my comments, I am willing to adjust my overall rating from a 3 to a 4.

---

> > > ### Author Response · Authors · 2025-08-05
> > >
> > > Dear reviewer fBa2,
> > >
> > > We sincerely appreciate your recognition of our rebuttal and insightful comment regarding the alignment between the user and item multimodal features $e^{mm}\_u$ and $e^{mm}\_v$.
> > >
> > > Due to the potentially large number of interactions per user, introducing explicit pairwise operations (e.g., dot-product) between every $ e^{mm}\_u $ and $ e^{mm}\_v $ during training would be computationally expensive and impractical.
> > > To address this, we revise our alignment loss to explicitly enforce the semantic consistency between the user multimodal representation $ e^{mm}\_u $ (generated via UEG) and the averaged representation derived directly from the historical interacted items' multimodal representations $ e^{mm}\_v $.
> > > The updated alignment loss is defined as:
> > >
> > > $$
> > > L\_{align}^{'} = \||e\_u^{id} - e\_u^{mm}\||^2\_2 + \lambda\||e\_u^{mm} - \frac{1}{s}\sum_{i=1}^s e\_{v\_i}^{mm}\||^2\_2.
> > > $$
> > >
> > > To verify the effectiveness of this modification, we conducted zero-shot experiments following the protocol of Appendix B.1 on the Baby dataset to evaluate the quality of $ e^{mm}\_u $ generated by the new model. We set $\lambda=0.05$, the table below illustrates that $ L\_{align}^{'} $ leads to better $ e^{mm}\_u $.
> > >
> > > | Generation Methods |Recall@20   | Recall@50  |
> > > |--------------------------|---------------|---------------|
> > > | Amazon                  |      0.0052      | 0.0150   |
> > > |     MLLM                 |         0.0140   | 0.0345   |
> > > |     w/o Lalign           |      0.0225      | 0.0426  |
> > > |     w $L_{align}$                  |    0.0276       | 0.0509    |
> > > |   w $L\_{align}^{'} $    |     **0.0284**    | **0.0522**     |
> > >
> > > On top of that, we use the revised model as a feature generator in AGM, and evaluated its AUC/ LogLoss performance on downstream recommendation tasks on the Baby dataset in the table below.
> > > | Model                    |AUC           | LogLoss  |
> > > |--------------------------|---------------|---------------|
> > > |    MLP(ID+MM) w $L_{align}$                  |  0.6719      | 0.6724    |
> > > | MLP(ID+MM)   w $L\_{align}^{'} $    |   0.6702      |  0.6786   |
> > > |    AGM w $L_{align}$                  |   0.6852     |  0.6683   |
> > > | AGM  w $L\_{align}^{'} $    |     0.6854    |  0.6678   |
> > > |    AGM* w $L_{align}$                  |  0.6864      |  0.6656   |
> > > | AGM*  w $L\_{align}^{'} $    |   0.6865      |  0.6656   |
> > >
> > >
> > > These experiments demonstrate the effectiveness of revised alignment loss $L_{align}^{'}$, which explicitly encourages $e_u^{mm}$ to reside in the same space as the item multimodal representations $e_v^{mm}$.
> > > $L_{align}^{'}$ reinforces semantic consistency across user and item multimodal embeddings.
> > > As shown in the zero-shot evaluation, this modification improves the quality of $e_u^{mm}$, achieving higher Recall compared to $L_{align}$. In downstream recommendation tasks, incorporating $L_{align}^{'}$ slightly maintains performance of AGM, confirming that the semantic benefits are realized without compromising recommendation effectiveness.
> > >
> > >
> > > Notably, while $L_{align}^{'}$ improves the semantic consistency of $e_u^{mm}$, its incorporation into the simple MLP (ID+MM) model leads to a drop performance.
> > > This can be attributed to the fact that MLP relies more heavily on the collaborative signals.
> > > Since $L_{align}^{'}$ additionally encourages $e_u^{mm}$ to align with the average multimodal embeddings of interacted items, it may slightly compromise the direct coordination between $e_u^{mm}$ and $e_u^{id}$.
> > > This introduces a trade-off between semantic alignment and discriminative power, which is more pronounced in lightweight architectures like MLP that lack sophisticated interaction modeling and rely heavily on the consistency between ID and multimodal representations.
> > >
> > > Finally, we sincerely appreciate your recognition of our work and the score increase. We are truly honored to know that our rebuttal has addressed your concerns. Once again, we sincerely appreciate your time and support. Wishing you all the best.

---

> > > > ### Comment · Reviewer_fBa2 · 2025-08-06
> > > >
> > > > Thank you for the detailed response. It has addressed my concerns.

---

> > > > > ### Author Response · Authors · 2025-08-06
> > > > >
> > > > > Dear Reviewer fBa2,
> > > > >
> > > > > Thank you for your response and for your positive evaluation. We are truly honored to know that our rebuttal has addressed your concerns. Once again, we sincerely appreciate your time and thoughtful feedback. Wishing you all the best.

---

### Official Review · Reviewer_a6WN · 2025-06-30

**Clarity:** 3
**Significance:** 2
**Originality:** 3
**Rating:** 4
**Confidence:** 2

**Summary:**

This paper proposes a two-stage framework for recommendation systems that first fine-tunes multimodal large language models (MLLMs) to generate unified user and item embeddings and then introduces Adaptive Gradient Masking (AGM) to dynamically balance gradient updates between ID-based and multimodal representations during joint training. The approach mitigates convergence inconsistency, leading to improved recommendation performance, as demonstrated by theoretical analysis and both offline and online experiments.

**Questions:**

1. AGM seems to need to calculate the Fisher Information Matrix and dynamic sampling mask. Will it cause unacceptable training/inference delays?
2. How does AGM perform when one type of information is of poor quality?

**Ethical Concerns:**

["NO or VERY MINOR ethics concerns only"]

**Final Justification:**

After reading the rebuttal and other reviews, I would like to maintain my score.

**Limitations:**

yes

**Quality:**

2

**Strengths And Weaknesses:**

Pros:
1. The author's motivation is clear and reasonable.
2. The proposed method corresponds intuitively to the motivation.
3. A theoretical analysis of the method is provided.

Cons：
1. Lack of time complexity analysis.
2. In the experiment, the author used AUC as the evaluation metric. Adding more recommendation-related evaluation metrics will help to keep consistent with the comparison method.
3. Regarding Section 4.4 Industrial Application, a more descriptive discussion would help to enhance the persuasiveness.

---

> ### Author Rebuttal · Authors · 2025-07-30
>
> Dear reviewer  a6WN,
>
> We sincerely thank you for your thoughtful feedback and positive score. We truly appreciate the time and effort you dedicated to evaluating our work. Below, we address each of your concerns in detail and provide clarifications where necessary.
>
> **Q1: Time complexity analysis**
>
> First, we clarify that our method adopts a two-stage framework, where the MLLM is fine-tuned offline to extract multimodal features prior to training the downstream model. Since this step is performed once and does not affect online training or inference, so it is reasonable to omit a detailed time complexity analysis for this stage.
>
> We provide a time complexity analysis of the AGM training process below:
>
> For a batch of size $|\mathcal{B}|$, the training time complexity of AGM consists of three components:
> 1.forward propagation: $\mathcal{O}(|\mathcal{B}|)$.
>
> 2.Fisher Information Matrix (FIM) computation: $\mathcal{O}(|\mathcal{B}| \cdot |\mathcal{P}|)$.  Here, $\mathcal{P}$ denotes the selected parameters for update, $\theta$ denotes the full parameter set, $|\mathcal{P}| \ll |\theta|$.
>
> 3.importance-based sampling via heap-based sorting: $\mathcal{O}(|\mathcal{P}| \log |\mathcal{P}|)$.
>
> Thus, we get the total training complexity per batch: $\boxed{\mathcal{O}(|\mathcal{B}| + |\mathcal{B}| \cdot |\mathcal{P}| + |\mathcal{P}| \log |\mathcal{P}|)}$.
>
> Although AGM introduces lightweight sampling and importance estimation overhead compared to standard training, its overall complexity remains near-linear with respect to the batch size. Therefore, AGM retains comparable scalability to standard models.
>
> **At inference stage, AGM incurs no additional complexity compared to conventional models, since neither FIM computation nor sampling is involved**. The inference complexity remains $\mathcal{O}(|\mathcal{B}|)$, consistent with standard recommendation architectures.
>
> **Q2: More recommendation-related evaluation metrics**
> In our current experiments, we adopt AUC and LogLoss as evaluation metrics. because our focus is on click-through rate (CTR) prediction, where these two metrics are widely used. We include Recall@10 and  Recall@20 results in the table below as a complementary evaluation:
>
> | Dataset     | Metric       | AlignRec | GUME   | AGM*   |
> |-------------|--------------|----------|--------|--------|
> | Baby        | Recall@10    | 0.0672    | 0.0671  | 0.0710  |
> |             | Recall@20    | 0.1047    | 0.1042  | 0.1069  |
> | Sports      | Recall@10    | 0.0758    | 0.0778  | 0.0813  |
> |             | Recall@20    | 0.1163    | 0.1166  | 0.1209  |
> | Electronics | Recall@10    | 0.0472    | 0.0458  | 0.0490  |
> |             | Recall@20    | 0.0699   | 0.0680  | 0.7260  |
>
> **Q3: More details of industrial application**
>
>
> We deploy our method in a large-scale short-video platform in China, which serves over 300 million DAUs. This application scenario involves high-throughput real-time recommendations across various traffic entry points. Given the immense volume and diversity of video content, real-time efficiency and embedding consistency are critical for system performance. Our proposed pipeline effectively addresses these challenges: by caching item-level multimodal embeddings extracted by the fine-tuned MLLM, we significantly reduce computational overhead during training and inference, enabling scalable deployment under stringent latency constraints. This architecture has been verified to support stable online serving and model updates on a weekly basis.
>
> **Q4: Robustness to low-quality multimodal representations**
>
> We agree that the quality of multimodal features can significantly influence recommendation performance. As discussed in Line 688 of the paper, the effectiveness of AGM depends in part on the quality and diversity of multimodal data.
>
> However, AGM is designed to remain robust even when one modality is degraded or missing. Specifically, the model includes an ID-based embedding branch and an alignment loss between multimodal and ID-based representations. This setup enables AGM to fall back on reliable ID features when multimodal features are noisy or incomplete, avoiding overfitting to poor-quality inputs.
>
> To further validate AGM’s robustness to low-quality multimodal features, we conducted ablation experiments using the three types of multimodal representations described in Table 6—(i) the base MLLM encoder without fine-tuning, (ii) our fine-tuned MLLM encoder, and (iii) our variant without the alignment loss—to evaluate the robustness of AGM to degraded or unaligned multimodal inputs.
>
> | Dataset | Metric     |  MLLM (i)     | w/o Lalign (iii)  |  AGM* (ii)  |
>  |---------|----------|----------|--------|--------|
>  |   Baby    | AUC        |  0.6743  | 0.6804  | 0.6864 |
>  |                | LogLoss  |   0.6696  |   0.6683 |  0.6656|
>  |   Sports    | AUC      |  0.6907  |   0.6981 |  0.7145|
>   |                | LogLoss  |  0.6450    |   0.6414 | 0.6391 |
>  |   Electronics | AUC  |  0.7082    |   0.7160 | 0.7310 |
>  |                | LogLoss  |   0.6076  |  0.5985 | 0.5969 |
>
> Finally, we truly appreciate all your valuable feedback. We hope our responses have addressed your concerns, and we would be grateful if you could take these clarifications into account in your overall evaluation. Thank you again for your time and insights — we look forward to any further comments you may have.

---

> > ### Comment · Reviewer_a6WN · 2025-08-05
> >
> > Thanks for the rebuttal. It has addressed part of my concerns. After reading the rebuttal and other reviews, I would like to maintain my positive score .

---

> > > ### Author Response · Authors · 2025-08-05
> > >
> > > Dear Reviewer a6WN,
> > >
> > > Thank you for your response and for recognizing our work. We are truly honored to know that our rebuttal has addressed your concerns. Once again, we sincerely appreciate your time and support. Wishing you all the best.

---

### Official Review · Reviewer_2Y2o · 2025-07-02

**Clarity:** 4
**Significance:** 3
**Originality:** 3
**Rating:** 5
**Confidence:** 4

**Summary:**

To address the convergence inconsistency problem in the joint training of ID-based and MLLM-based representations in recommendation systems, the authors propose a two-stage framework: (1) fine-tuning MLLMs to generate unified user/item representations while aligning them with ID features, and (2) introducing an Adaptive Gradient Masking (AGM) strategy to dynamically balance parameter updates between ID and multimodal branches during joint training. The method is evaluated through extensive offline experiments and online A/B tests, demonstrating effectiveness in mitigating convergence issues and improving recommendation performance.

**Questions:**

1) Could you provide more details about the computational resources required for MLLM fine-tuning? This would help assess practical deployment feasibility.
2) Have you considered testing your method on recommendation domains beyond e-commerce/short videos (e.g., music or news recommendation) to demonstrate broader applicability?
3) The paper mentions AGM* improves upon AGM's bias - could you quantify how much performance gain comes specifically from this unbiased variant?

**Ethical Concerns:**

["NO or VERY MINOR ethics concerns only"]

**Final Justification:**

The authors addressed all my concerns from Weaknesses and Questions, so I lifted my final score.

**Limitations:**

Yes

**Quality:**

3

**Strengths And Weaknesses:**

Strengths
1) The proposed AGM mechanism is innovative, using mutual information to estimate representation contributions and applying non-uniform gradient masking at the sub-network level. The theoretical analysis of AGM's convergence properties adds rigor to the approach.
2) Comprehensive experimental validation across multiple datasets and backbone architectures demonstrates the method's robustness. The online A/B test results showing significant improvements are particularly convincing for real-world applicability.
3) The paper provides thorough ablation studies (Table 3) analyzing each component's contribution, and includes a zero-shot evaluation (Table 6) to assess multimodal representation quality, both of which strengthen the methodological claims.

Weaknesses
1) The computational cost of fine-tuning MLLMs for recommendation tasks isn't adequately discussed. Given the large parameter size of MLLMs, this could be a practical limitation for some applications.
2) While multiple datasets are used, they're all from similar domains (e-commerce and short videos). Testing on more diverse recommendation scenarios (e.g., news, music) would better demonstrate generalizability.
3) The comparison with baseline methods (Table 2) could be enhanced by including more recent MLLM-based recommendation approaches beyond those cited.

---

> ### Author Rebuttal · Authors · 2025-07-30
>
> Dear Reviewer 2Y2o,
>
> We sincerely thank you for your thoughtful feedback and positive score. We truly appreciate the time and effort you dedicated to evaluating our work. Below, we address each of your concerns in detail and provide clarifications where necessary.
>
> **Q1: Computational cost of fine-tuning**
>
> We thank the reviewer for raising the concern regarding the computational cost of fine-tuning large MLLMs. As noted in Lines 575–576 of our paper, we adopt Qwen2vl-2b as the backbone model and fine-tune it for 5 epochs across four datasets, using a batch size of 128 on 4 A100 GPUs. This provides a concrete reference for the resource requirements in our setting.
>
> Additionally, we emphasize that our method adopts a two-stage franework: the MLLM is fine-tuned offline in the first stage to extract high-quality multimodal embeddings, while the downstream recommendation model is trained separately in the second stage. For practical deployment, we cache all item multimodal embeddings, thus eliminating the need for repeated MLLM inference during training or inference. As a result, the computational burden of fine-tuning is incurred only once offline and has minimal impact on online efficiency.
>
> **Q2: Evaluation of more datasets**
>
> To further evaluate the generalizability of our method beyond e-commerce and short video domains, we include the MIND-small dataset, a widely-used benchmark for news recommendation.
>
> The table below summarizes the performance of AGM*, AlignRec and GUME on MIND-small:
>
> | Metric   | AlignRec | GUME   | AGM*   |
> |----------|----------|--------|--------|
> | AUC      | 0.6823   | 0.6835 | 0.6862 |
> | LogLoss  | 0.5934  | 0.5902 | 0.5857 |
>
> **Q3: Comparison with more MLLM methods**
>
> We additionally include LIRDRec [1] as a baseline in our evaluation. LIRDRec [1] is designed to learn item representations directly from multimodal content. The following table summarizes the Recall@10 and Recall@20 results across three datasets, which illustrates that AGM* performs on par with or slightly better than LIRDRec and other baselines.
>
> | Dataset     | Metric       | AlignRec | LIRDRec | GUME   | AGM*   |
> |-------------|--------------|----------|---------|--------|--------|
> | Baby        | Recall@10    | 0.0672   | 0.0707  | 0.0671 | 0.0710 |
> |             | Recall@20    | 0.1047   | 0.1068  | 0.1042 | 0.1069 |
> | Sports      | Recall@10    | 0.0758   | 0.0815  | 0.0778 | 0.0813 |
> |             | Recall@20    | 0.1163   | 0.1216  | 0.1166 | 0.1209 |
> | Electronics | Recall@10    | 0.0472   | 0.0493  | 0.0458 | 0.0490 |
> |             | Recall@20    | 0.0699   | 0.0736  | 0.0680 | 0.0742 |
>
> **Q4: Quantification of AGM\*'s improvement beyond AGM**
>
> We provide a clearer quantification of AGM\*’s improvement over AGM in Table 2. For example, on the Amazon Baby dataset, AUC increases from 0.6852 (AGM) to 0.6864 (AGM\*), and LogLoss decreases from 0.6683 (AGM) to 0.6656 (AGM\*). Similar gains are observed across other datasets. We apologize that Table 2 may not have illustrated this improvement clearly enough. In the revised version, we will include a delta row in the table to highlight AGM\*’s performance gains more explicitly.
>
> Finally, we truly appreciate all your valuable feedback. We hope our responses have addressed your concerns, and we would be grateful if you could take these clarifications into account in your overall evaluation. Thank you again for your time and insights — we look forward to any further comments you may have.
>
> [1] Learning Item Representations Directly from Multimodal Features for Effective Recommendation

---

> ### Comment · Reviewer_2Y2o · 2025-08-05
>
> Thank you very much for your response. I have read your response to my comments and other reviewers. The authors address my concerns.

---

> > ### Author Response · Authors · 2025-08-05
> >
> > Dear Reviewer 2Y2o,
> >
> > Thank you for your response and for your positive evaluation. We are truly honored to know that our rebuttal has addressed your concerns. Once again, we sincerely appreciate your time and thoughtful feedback. Wishing you all the best.

---

### Official Review · Reviewer_QngB · 2025-07-04

**Clarity:** 3
**Significance:** 4
**Originality:** 3
**Rating:** 4
**Confidence:** 5

**Summary:**

The paper addresses the convergence inconsistency in large-scale recommendation systems by proposing a two-stage framework involving multimodal representation learning and joint training optimizations. The authors also provide theoretical analysis on the convergence properties of the proposed method.  Both offline and online experiments validate the proposed AGM strategy.

**Questions:**

1. Table 2 presents the overall recommendation performance, where the proposed method shows an advantage over GUME, though the margin is relatively small. This raises the question of how the proposed method compares to long-tail-optimized methods like GUME specifically on long-tail items.

2. Table 6 shows the evaluation of multimodal representations, which demonstrates the proposed method outperforms the "MLLM" method with a large margin. But I am concerned about the fairness of the evaluation. If I understand correctly, AGM fine-tunes MLLM with user behavior, which could be a major contributing factor to its performance under the Intermediate Evaluation Protocols of AlignRec.

**Ethical Concerns:**

["NO or VERY MINOR ethics concerns only"]

**Final Justification:**

The authors’ rebuttal has successfully addressed all of my concerns. I have also reviewed the discussions between the authors and other reviewers. Based on the clarifications provided and the overall discourse, I believe that a borderline accept is a reasonable and fair recommendation.

**Limitations:**

yes

**Paper Formatting Concerns:**

None.

**Quality:**

3

**Strengths And Weaknesses:**

Strengths

1. The motivation of the proposed framework is convincing.

2. The proposed AGM strategy is innovative. It uses mutual information to estimate representation contributions and apply non-uniform gradient masking to implement modality-specific gradient updates.

3. The theoretical analysis of AGM adds rigor to the work.

4. The experiments demonstrate significant performance across different categories of datasets. The inclusion of online A/B testing results also demonstrates the practical effectiveness of the proposed method in real-world industrial applications.

Weaknesses

1. Although the authors provide implementation code in the supplementary material, it appears that the implementation of the proposed multimodal representation learning method is missing. Including this part would enhance the reproducibility and credibility of the work.

2. In Lines 23-24, the paper mentions that “they suffer from limited generalization, making them inadequate for addressing long-tail items, data sparsity, and cold-start scenarios.” However, it appears that the experimental evaluation does not include these scenarios. Including such evaluations would strengthen the empirical validation of the proposed method.

---

> ### Author Rebuttal · Authors · 2025-07-30
>
> Dear reviwer QngB,
>
> We sincerely thank you for your thoughtful feedback and positive score. We truly appreciate the time and effort you dedicated to evaluating our work. Below, we address each of your concerns in detail and provide clarifications where necessary.
>
> **Q1: Supplement code**
>
> Due to the NeurIPS 2025 official policy, updating the submitted github repository is not permitted during the rebuttal phase. We will update the repository to include the missing multimodal representation learning module immediately after the rebuttal process.
>
> **Q2: Long-tail evaluation**
>
> We have added additional experiments to directly evaluate our method’s performance on long-tail items. Specifically, for each dataset, we sort items by their total number of user interactions in descending order. We then partition the items into the following frequency bands:
> - High-frequency: top 20% most popular items.
> - Mid-frequency: items ranked in the 20%–50% range.
> - Low-frequency: the remaining 50% of items.
>
> We present the results in the table below. We select GUME as the primary baseline and conduct evaluations on three datasets. The results show that AGM achieves significant performance improvements on low-frequency items, demonstrating its effectiveness in addressing the long-tail challenge.
>
> | Dataset     | Frequency  | AUC (GUME) | AUC (AGM*) | LogLoss (GUME) | LogLoss (AGM*) |
> |-------------|------------|------------|------------|----------------|----------------|
> | Baby        | Overall    | 0.6832     | 0.6864     | 0.6680         | 0.6656         |
> |             | High Freq  | 0.6861     | 0.6873     | 0.6562         | 0.6544         |
> |             | Mid Freq   | 0.6810     | 0.6847     | 0.6793         | 0.6760         |
> |             | Low Freq   | 0.6752     | 0.6801     | 0.6960         | 0.6936         |
> | Sports      | Overall    | 0.7119     | 0.7145     | 0.6399         | 0.6391         |
> |             | High Freq  | 0.7182     | 0.7211     | 0.6201         | 0.6186         |
> |             | Mid Freq   | 0.7065     | 0.7090     | 0.6575         | 0.6556         |
> |             | Low Freq   | 0.6902     | 0.7010     | 0.6842         | 0.6823         |
> | Elec.       | Overall    | 0.7271     | 0.7310     | 0.5991         | 0.5969         |
> |             | High Freq  | 0.7295     | 0.7333     | 0.5904         | 0.5886         |
> |             | Mid Freq   | 0.7231     | 0.7255     | 0.6122         | 0.6103         |
> |             | Low Freq   | 0.7110     | 0.7169     | 0.6235         | 0.6210         |
>
> **Q3: Performance improvement seems relatively small**
>
> In the CTR predictions, an improvement as small as 0.001 in AUC is generally considered significant, as even minor gains in offline metrics can lead to substantial performance boosts in online serving scenarios [1]. For example, in [2], the Wide & Deep model achieved only a 0.275% AUC improvement over logistic regression in offline evaluation, but led to a 3.9% increase in online CTR.
>
> **Q4: Fairness of the evaluation**
>
> We sincerely thank you for your thoughtful question. We understand the concern regarding the fairness of the evaluation and offer the following clarifications:
> - Our evaluation protocols follow AlignRec exactly. We clarify that the evaluation protocols used in table 6 follow AlignRec exactly, in both the dataset split and the Intermediate Evaluation Protocol setup. Consistent with AlignRec, we compare the representation quality of different multimodal encoders under the same user behavior-based evaluation.
>
> - AlignRec itself also fine-tunes its MLLM. Different models adopt different strategies for using multimodal features. AlignRec itself includes a fine-tuned MLLM encoder (MMEnc) as part of its pipeline. Similarly, in Table 6, we report results from (i) the base frozen MLLM encoder, (ii) our fine-tuned version, and (iii) our model without the alignment loss. AlignRec’s Table 3 makes an analogous comparison between (i) Amazon, (ii) CLIP, and (iii) their MMEnc, which also shows significant gains from fine-tuning.
>
> - The key difference lies in the fine-tuning strategy employed. Given that both methods generate comparable and consistent multimodal embeddings, the performance gains achieved by AGM over AlignRec can be attributed to the effectiveness of our subsequent representation learning and aggregation mechanisms. Table 2 shows that AGM still outperforms AlignRec, indicating that our performance gains cannot be solely attributed to MLLM fine-tuning. Rather, they result from AGM’s ability to dynamically integrate ID and multimodal features.
>
> Finally, we truly appreciate all your valuable feedback. We hope our responses have addressed your concerns, and we would be grateful if you could take these clarifications into account in your overall evaluation. Thank you again for your time and insights — we look forward to any further comments you may have.
>
> [1] Guo H, Tang R, Ye Y, et al. DeepFM: a factorization-machine based neural network for CTR prediction[J]. arXiv preprint arXiv:1703.04247, 2017.
>
> [2] Cheng H T, Koc L, Harmsen J, et al. Wide & deep learning for recommender systems[C]//Proceedings of the 1st workshop on deep learning for recommender systems. 2016: 7-10.

---

> > ### Comment · Reviewer_QngB · 2025-08-06
> >
> > Thanks for the detailed reply! That really helped clear things up.

---

> > > ### Author Response · Authors · 2025-08-06
> > >
> > > Dear Reviewer QngB,
> > >
> > > Thank you for your response and for your positive evaluation. We are truly honored to know that our rebuttal has addressed your concerns. Once again, we sincerely appreciate your time and thoughtful feedback. Wishing you all the best.

---

### Decision · Program_Chairs · 2025-09-17

**Decision:**

Accept (poster)

**Comment:**

This paper introduces the Adaptive Gradient Masking technique for recommendation tasks. Initially, the reviewers raised concerns regarding the lack of implementation details, insufficient experimental results, and the computational cost of fine-tuning MLLMs. However, after the rebuttal, the authors successfully addressed most of these concerns, including clarifications of novelty, and all reviewers provided positive feedback. The AC has carefully reviewed the paper, the reviewer comments, and the rebuttal, and agrees that the paper is well-motivated, clearly written, and supported by thorough experiments. Therefore, the AC recommends acceptance. It is encouraged that the authors incorporate all additional experiments and discussions from the rebuttal into the final version.